# Scalable and Cost-Efficient de Novo Template-Based Molecular Generation

**Piotr Gaiński**[*,1]**, Oussama Boussif**[*,2,3]**, Andrei Rekesh**[4]**, Dmytro Shevchuk**[5,4]**, Ali Parviz**[2,3]**,
**Mike Tyers**[5,4]**, Robert A. Batey**[4,6]**, Michał Koziarski**[5,4,7,6]
[1] Jagiellonian University, Faculty of Mathematics and Computer Science,
[2] Mila – Québec AI Institute, [3] Université de Montréal,
[4] University of Toronto, [5] The Hospital for Sick Children Research Institute,
[6] Acceleration Consortium, [7] Vector Institute, [*] Equal contribution
`piotr.gainski@doctoral.uj.edu.pl`

## Abstract

Template-based molecular generation offers a promising avenue for drug design by ensuring generated compounds are synthetically accessible through predefined reaction templates and building blocks. In this work, we tackle three core challenges in template-based GFlowNets: (1) minimizing synthesis cost, (2) scaling to large building block libraries, and (3) effectively utilizing small fragment sets. We propose **Recursive Cost Guidance**, a backward policy framework that employs auxiliary machine learning models to approximate synthesis cost and viability. This guidance steers generation toward low-cost synthesis pathways, significantly enhancing cost-efficiency, molecular diversity, and quality, especially when paired with an **Exploitation Penalty** that balances the trade-off between exploration and exploitation. To enhance performance in smaller building block libraries, we develop a **Dynamic Library** mechanism that reuses intermediate high-reward states to construct full synthesis trees. Our approach establishes state-of-the-art results in template-based molecular generation.

## 1 Introduction

Generative models hold significant promise for accelerating drug discovery by enabling direct sampling of molecules with desired properties, vastly expanding the accessible chemical space. However, despite increasing interest from the machine learning community, their adoption in practical screening pipelines remains limited, primarily due to the poor synthesizability of the generated compounds. Recent research has begun to address this synthesizability bottleneck.

One prominent direction involves directly optimizing synthesizability, either by including synthesizability scores as part of the reward function [1], or by using retrosynthesis models to guide goal-conditioned generation [2]. However, such approaches remain fundamentally unreliable: heuristic metrics such as SAScore [3] often correlate poorly with true synthetic feasibility, while learned models such as RetroGNN [4] struggle with limited training data and poor generalization.

An increasingly popular and more principled alternative is template-based synthesis, where molecules are assembled from predefined building blocks through well-characterized reactions [5, 6]. This paradigm mirrors laboratory synthesis more closely and, with appropriate reaction templates, can provide an approximate guarantee of synthesizability. Recent work has embedded this formulation within Generative Flow Networks (GFlowNets) [7], which are well suited to molecular discovery due to their capacity for mode-seeking sampling and efficient exploration of combinatorial spaces [8–10].

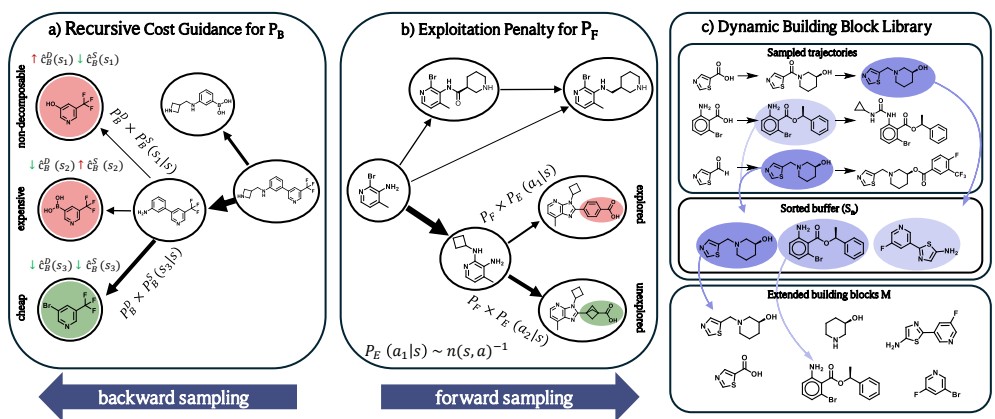

Figure 1: **a)** Recursive Cost Guidance employs machine learning models $\hat{c}_B^S$ and $\hat{c}_B^D$ to estimate the intractable synthesis cost and decomposability of precursor molecules, guiding the backward policy $P_B$ toward cheaper and viable intermediates. **b)** Forward policy is augmented with Exploitation Penalty to counter overexploitation induced by Synthesis Cost Guidance component of $P_B$. **c)** Dynamic Library gathers intermediate molecules with the highest expected reward and adds them to the building block library $M$, enabling full-tree synthesis.

Despite this progress, key limitations remain, particularly in terms of scalability and cost awareness. An ideal synthesis-aware generative model would explore the full space of synthesizable molecules while favoring those that are simpler and cheaper to make. Existing methods fall short: RGFN [8] employs a small, curated fragment library optimized for wet lab feasibility, but this restricts scalability. In contrast, RxnFlow [10] improves scalability through larger libraries and broader exploration, but ignores the cost of the synthesis, reducing its practical utility in downstream screening campaigns.

In this paper, we propose SCENT (**S**calable and **C**ost-**E**fficient de **N**ovo **T**emplate-Based Molecular Generation) which addresses the aforementioned limitations of reaction-based GFlowNets. Our contributions can be summarized as follows (graphical summary in Figure 1):

- We introduce **Recursive Cost Guidance** for the backward policy, which utilizes a machine learning model to approximate the recursive cost of backward transitions. From this, we derive two targeted strategies: **Synthesis Cost Guidance**, which reduces synthesis cost and improves diversity in large building block settings; and **Decomposability Guidance**, which enforces the validity of backward transitions by discouraging transitions through indecomposable molecules.

- We analyze the exploitative behavior induced by **Synthesis Cost Guidance** and propose **Exploitation Penalty**, a simple yet effective regularizer that promotes exploration by penalizing repeated state-action pairs. This improves performance across all benchmarks.

- We develop **Dynamic Library** mechanism that augments the initial set of building blocks with high-reward intermediates discovered during optimization. This mechanism enables full-tree synthesis and increases mode coverage in both low- and high-resource scenarios.

- We benchmark recent reaction-based GFlowNets across three library sizes and three design tasks, demonstrating that SCENT significantly outperforms prior methods in terms of synthesis cost, molecular diversity, and the number of high-reward compounds discovered.

Our code is publicly available at `https://github.com/koziarskilab/SCENT`.

## 2 Related Works

**Molecular Generation.** A broad range of machine learning approaches has been proposed for small molecule generation, varying in both molecular representations—such as graphs, SMILES, or 3D coordinates—and generative paradigms, including diffusion models [11–14], flow matching [15–17], reinforcement learning [18–20], and large language models (LLMs) [21–24]. Among these, Generative Flow Networks (GFlowNets) [7, 25–28] have emerged as a compelling alternative to

reinforcement learning for iterative molecular generation. Their ability to perform diverse, mode-covering sampling makes them particularly well suited for drug discovery.

**Synthesis-Aware Molecular Generation.** Synthesis-aware molecular generation has gained momentum as a strategy to ensure that generated molecules are not only property-optimized but also synthetically accessible. These methods construct molecules from predefined building blocks using high-yield, well-characterized reactions. Recent GFlowNet-based approaches [9, 8, 10], along with reinforcement learning [5, 6], genetic algorithms [29], and autoregressive models [30] incorporate reaction-aware priors to produce more realistic candidates.

**Backward Policy Design in GFlowNets.** In Generative Flow Networks (GFlowNets), the most common training objective is to match the forward policy $P_F$ and the backward policy $P_B$ to satisfy the trajectory balance condition [31]. While $P_B$ is typically learned jointly with $P_F$, it can also be optimized separately or fixed to influence the behavior of $P_F$. For example, Jang et al. [32] train $P_B$ via maximum likelihood over observed trajectories to induce exploitative behavior in $P_F$, whereas Mohammadpour et al. [33] use a fixed maximum-entropy $P_B$ to encourage exploration.

Guided Trajectory Balance (GTB) [34] is the most relevant prior work on backward policy guidance. GTB introduces a preference distribution over trajectories to a given terminal state and trains $P_B$ to match it. In contrast, we define trajectory preference through a recursive cost function, yielding a more interpretable and flexible formulation. We approximate this cost using an auxiliary model, which gives rise to our proposed **Recursive Cost Guidance** framework that steers generation toward low-cost synthesis pathways while improving consistency with the underlying Markov decision process.

SynFlowNet [9] also modifies $P_B$, using reinforcement learning to discourage transitions through nondecomposable intermediates. While it addresses similar challenges, our method provides a more general and principled solution by unifying synthesis cost reduction and MDP-consistent trajectory shaping through backward policy design.

**Library Learning.** Library learning is a long-standing strategy for managing large search spaces in domains such as program induction and reinforcement learning. DreamCoder [35–37], for example, extracts reusable subroutines from sampled programs to build a library of primitives. Similarly, the Options framework [38] and macro-actions [39] learn temporally extended actions to facilitate exploration and improve credit assignment. More recently, ActionPiece [40] applied byte pair encoding (BPE) to compress frequent action subsequences in GFlowNet trajectories, augmenting the action space with learned macro-actions.

Our **Dynamic Library** builds on this foundation but shifts from **action abstractions** to **state abstractions**. Rather than compressing action sequences, we identify and cache high-reward intermediate states, which act as reusable substructures for constructing nonlinear synthesis trees. While conceptually similar to DreamCoder's structural reuse, our approach operates in the space of synthesis pathways, a novel application of library learning principles to generative models for molecular design.

## 3 Method

In this section, we present SCENT and its core components: (1) Recursive Cost Guidance, a general backward policy framework that estimates recursive transition costs using a learned model; (1.1) Synthesis Cost Guidance, which biases generation toward low-cost synthesis pathways; (1.2) Decomposability Guidance, which discourages selection of indecomposable molecules; (2) Exploitation Penalty, which counterbalances the exploitative bias introduced by Synthesis Cost Guidance (see Section 4.4) and improves overall performance; and (3) Dynamic Building Block Library, which expands the building block set with high-reward intermediates to enable full-tree synthesis and trajectory compression. The complete pseudocode of SCENT's training procedure is provided in Appendix A.1.

### 3.1 Preliminaries

Like other reaction-based GFlowNets [8–10], SCENT operates on a predefined set of reaction templates $R$ and building blocks $M$. Its forward policy $P_F$, adapted from RGFN (Appendix A.2),

constructs molecules via sequential application of reactions. The backward policy $P_B$, in turn, decomposes a molecule by reversing these reactions. Specifically, given an intermediate molecule $m$, $P_B$ samples a tuple $(m', M', r)$ where $m'$ is a precursor molecule, $M'$ is a tuple of building blocks from $M$, and $r \in R$ is a reaction such that applying $r$ to $m'$ and $M'$ yields $m$. For clarity, we denote molecules $m$ and $m'$ as states $s$ and $s'$, and the pair $(M', r)$ as an action $a'$, so that the backward policy is defined as $P_B((s', a') \mid s)$.

The backward policy in SCENT is constructed using the **Recursive Cost Guidance** framework introduced in Section 3.2. It is composed of two sub-policies: the **Synthesis Cost Guidance** policy $P_B^S$ (Section 3.2.1) and the **Decomposability Guidance** policy $P_B^D$ (Section 3.2.2).

## 3.2 Recursive Cost Guidance

Recursive Cost Guidance is a new and general framework for backward policy guidance that employs a machine learning model to approximate the recursive costs of backward transitions. Let $c_F(\tau_{0:n})$ be a function that assigns a cost to a forward trajectory $\tau_{0:n} = (s_0, a_0, \ldots, s_n)$ starting from source state $s_0$:

$$c_F(\tau_{0:n}) = C(c_F(\tau_{0:n-1}), s_{n-1}, a_{n-1}, s_n), \tag{1}$$

where $C$ defines the cost of transition from $s_{n-1}$ to $s_n$ via action $a_{n-1}$, possibly depending on the cumulative cost $c_F(\tau_{0:n-1})$. A natural instantiation of $c_F$ for template-based GFlowNets is the cost of the synthesis pathway, described in Section 3.2.1. To guide the backward policy $P_B$ toward trajectories with lower forward cost $c_F$, we define a backward cost function $c_B$ that represents the cost of the cheapest forward trajectory reaching a given state $s$:

$$c_B(s) = \min_{\tau \in \mathcal{T}_{\to s}} c_F(\tau), \tag{2}$$

where $\mathcal{T}_{\to s}$ is the set of all trajectories leading to state $s$. This can be reformulated into:

$$c_B(s) = \min_{(s', a') \in SA'} C(c_B(s'), s', a', s), \tag{3}$$

where $SA'$ is a set of all parent state-action tuples. Since $c_B$ is defined *recursively*, exact computation is generally *intractable*. To address this, we estimate $c_B$ using a machine learning model $\hat{c}_B$, and define recursively guided backward policy $P_B$:

$$P_B((s_i', a_i')|s) = \sigma_i^{|SA'|}(\mathbf{l}), \; l_i = -\alpha C(\hat{c}_B(s_i'), s_i', a_i', s), \tag{4}$$

where $\alpha$ controls the greediness, and $\mathbf{l}$ is the vector of logits $l_i$. This formulation avoids expensive recursive evaluations of $c_B$ by leveraging an inexpensive surrogate $\hat{c}_B$ (see Figure 2). To train $\hat{c}_B$, we maintain a fixed-size dataset $D$ containing current estimates of $c_B(s)$ denoted $\tilde{c}_B(s)$. After each iteration of forward policy training, we update these estimates using the encountered trajectories $\tau_{0:n}$ by applying: $\tilde{c}_B'(s_i) = \min(\tilde{c}_B'(s_i), c_F(\tau_{0:i}))$. The model $\hat{c}_B$ is then fitted to $D$ using a dedicated cost loss $\mathcal{L}_{cost}$. In this work, we design two specific cost models — one for **Synthesis Cost Guidance** and another for **Decomposability Guidance**. The details of the models can be found in Appendix A.3.

### 3.2.1 Synthesis Cost Guidance

Even though template-based approaches, if used with a curated set of reaction templates and fragments, can guarantee high synthesizability, the cost of synthesis of different molecules can vary widely depending on the cost of reactants and the yields of reactions. RGFN [8] approaches this issue by defining a highly curated set of low-cost fragments and high-yield reactions. However, this comes at the price of reducing molecular diversity and reachable chemical space. Instead, we propose defining a more extensive fragment library, but guiding the generation process toward cheaper molecules.

We estimate synthesis cost using stock prices of building blocks and reaction yields:

$$C^S(x, s', a', s) = (x + c(a'))y(s', a', s)^{-1}, \tag{5}$$

where $x$ represents either the accumulated cost of a synthesis pathway (in Equation (6)) or the minimal cost to synthesize a molecule (in Equation (7)). The term $c(a')$ denotes the total cost of all reactants involved in the action $a'$ ($a'$ includes the building blocks $M'$ and the reaction template $r$). The reaction yield $y(s', a', s)$ is the expected success rate of the reaction leading from reactants $\{s'\} \cup M'$ to the product $s$ via the template $r$. Details on how yields and building block costs are

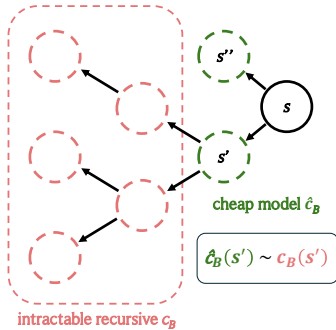
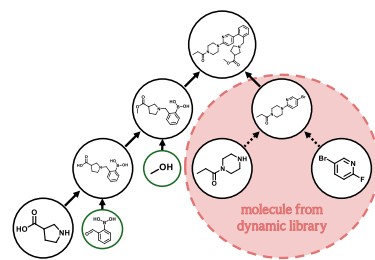

Figure 2: Recursive Cost Guidance framework uses a cheap model $\hat{c}_B$ to approximate the intractable recursive cost $c_B$ of backward transitions.

Figure 3: Dynamic building block library is augmented using high-reward molecules (depicted in pink) that occur during sampling.

estimated are provided in Appendix E. While other factors, such as reaction time or solvent cost, may influence synthesis cost, they are either highly lab-specific or typically negligible. Consequently, our formulation of $C^S$ offers a robust and broadly applicable first-order approximation.

We integrate $C^S$ into the forward and backward cost formulations (Equation (1) and 3) as follows:

$$c_F^S(\tau_{0:n}) = C^S(c_F^S(\tau_{0:n-1}), s_{n-1}, a_{n-1}, s_n), \tag{6}$$

$$c_B^S(s) = \min_{(s',a') \in SA'} C^S(c_B^S(s'), s', a', s), \tag{7}$$

As in Section 3.2, we use a machine learning model $\hat{c}_B^S$ to approximate the intractable $c_B^S$:

$$P_B^S((s_i', a_i')|s) = \sigma_i^{|SA'|}(\mathbf{1}), \ l_i = -\alpha_S C^S(\hat{c}_B^S(s_i'), s_i', a_i', s), \tag{8}$$

Further architectural and training details for $\hat{c}_B^S$ are provided in Appendix A.3. Our experiments show that the use of $P_B^S$ significantly reduces the cost of the proposed synthesis routes while dramatically increasing the number of distinct modes discovered (Section 4).

### 3.2.2 Decomposability Guidance

A core challenge in template-based GFlowNets is ensuring that the parent state $s'$, sampled by the backward policy $P_B((s', a') \mid s)$, is recursively decomposable into building blocks. RGFN addresses this by explicitly verifying decomposability, but this incurs significant computational cost [8]. In contrast, Cretu et al. [9] propose two implicit strategies: (1) a reinforcement learning approach that assigns positive rewards to trajectories reaching the source state $s_0$ and negative rewards to those terminating in non-decomposable intermediates; and (2) a pessimistic backward policy trained via maximum likelihood over forward-sampled trajectories to implicitly favor valid paths.

In this work, we take a different approach by leveraging our Recursive Cost Guidance framework to address decomposability. Specifically, we define a backward cost function that penalizes invalid trajectories that do not begin at a source state:

$$c_B^D(s) = \min_{(s',a') \in SA'} c_B^D(s'), \tag{9}$$

with the base case $c_B^D(s) = 0$ if $s$ is a source state and $SA' = \emptyset$, and $c_B^D(s) = 1$ otherwise. We then define the backward policy $P_B^D$, leveraging $\hat{c}_B^D$ model that approximates the intractable cost $c_B^D$:

$$P_B^D((s_i', a_i') \mid s) = \sigma_i^{|SA'|}(\mathbf{1}), \quad l_i = -\alpha_D \hat{c}_B^D(s_i'), \tag{10}$$

Further architectural and training details for $\hat{c}_B^D$ are provided in Appendix A.3. Decomposability Guidance is a critical component of SCENT, substantially influencing overall performance (see Appendix J.1). Unless otherwise noted, all experiments in this paper assume that SCENT uses Decomposability Guidance. When Synthesis Cost Guidance is also enabled, SCENT combines the two by taking the product of their corresponding backward policies:

$$P_B((s', a') \mid s) \propto P_B^C((s', a') \mid s) \cdot P_B^D((s', a') \mid s). \tag{11}$$

### 3.3 Exploitation Penalty

As shown in Section 4.4, incorporating Synthesis Cost Guidance into SCENT induces more exploitative behavior. To counteract this tendency, we introduce a simple yet effective strategy that penalizes repeated action choices in a given state. Specifically, we augment the forward policy $P_F$ with an exploration-aware component $P_E$:

$$P'_F(a \mid s) \propto P_F(a \mid s) \cdot P_E(a \mid s), \quad P_E(a \mid s) = \frac{(n(s,a) + \epsilon)^{-\gamma}}{\sum_{a'} (n(s,a') + \epsilon)^{-\gamma}}. \tag{12}$$

Here, $n(s,a)$ denotes the number of times action $a$ has been selected at state $s$, $\epsilon > 0$ is a smoothing term, and $\gamma > 0$ controls the strength of the penalty.

Unlike pseudo-count methods that rely on approximate density models [41–43], our method uses exact counts, avoiding estimation artifacts and simplifying implementation. Although this count-based approach requires explicit storage of visited state-action pairs—which could be problematic in large state spaces—we show in Section 4.4 that applying this penalty only during the initial training phase (e.g., the first 1,000 iterations) is sufficient to yield consistent exploration benefits. This keeps the memory footprint bounded and independent of the total number of training iterations.

### 3.4 Dynamic Building Block Library

Current template-based GFlowNets generate only left-leaning synthesis trees by drawing reactants from a fixed set of building blocks (Figure 3). We extend this setup with a Dynamic Building Block Library that augments the block set with high-reward intermediates discovered during training. These intermediates act as reusable subtrees, enabling the construction of richer, tree-structured synthesis pathways. This not only expands the space of reachable molecules (more down in the section) but also allows trajectory compression (rewriting paths into compact trees), enhancing credit assignment and training efficiency.

Every $T$ training iterations, intermediate molecules encountered during the training are ranked by their expected utility $U(s)$:

$$U(s) = \frac{1}{|\mathcal{T}_s|} \sum_{\tau \in \mathcal{T}_s} R(s_\tau), \tag{13}$$

where $\mathcal{T}_s$ is the set of trajectories containing molecule $s$, and $R(s_\tau)$ is the reward assigned by GFlowNet to the final state of the trajectory $\tau$. The top-$L$ intermediates, compatible with known reaction patterns, are added to the library. This procedure is repeated up to $N_{\text{add}}$ times. This approach significantly improves the performance of SCENT in all tested configurations (see Section 4).

To assess the extent to which Dynamic Library expands the space of reachable molecules, we randomly generated 10,000 balanced binary synthesis trees (emulating Dynamic Library) and attempted to decompose each resulting molecule into synthesis paths accessible to SCENT without Dynamic Library. We found that 43% of the synthesis trees could not be decomposed, indicating that Dynamic Library increases the number of reachable molecules by up to 75%.

## 4 Experiments

In Section 4.2, we compare SCENT against other template-based GFlowNets and show the contribution of individual SCENT's components to overall performance. Section 4.3 analyzes the cost reduction mechanism behind the Synthesis Cost Guidance, Section 4.4 highlights its inherent exploitative behavior, motivating the Exploitation Penalty, and Section 4.5 demonstrates the scaling properties of the Synthesis Cost Guidance. Finally, Section 4.6 shows that SCENT displays improved synthesizability compared to other approaches.

### 4.1 Setup

Previously reported results for template-based GFlowNets are not directly comparable due to differences in template and building block sets. To ensure fairness, we introduce a unified benchmark with standardized templates and building blocks. We evaluated models in three settings: (1) **SMALL**:

curated fragments (see Appendix E) for rapid synthesis or high-throughput screening, (2) **MEDIUM**: SynFlowNet's templates with 64k Enamine building blocks, and (3) **LARGE**: like MEDIUM, but with 128k building blocks. These settings span practical use cases: SMALL focused on rapid synthesis and MEDIUM/LARGE supporting broader exploration requiring external procurement.

The models are tested in the sEH proxy task [7], along with the GSK3$\beta$ and JNK3 tasks [44–47] (see Appendix F for results on these tasks). We report metrics for molecules discovered during training, including: (1) the number of high-reward molecules with pairwise ECFP6 Tanimoto similarity below 0.5, and (2) the number of unique Bemis-Murcko scaffolds among high-reward molecules (where high-reward is defined by a threshold). Additionally, we sample 1,000 synthesis paths and report: (1) the average reward of resulting molecules, and (2) the average synthesis cost of high-reward paths. Full training details and hyperparameters are in Appendix B.

## 4.2   Evaluation of Template-Based Generative Models

In this section, we evaluate SCENT against recent template-based GFlowNet models: RGFN [8], SynFlowNet [9], and RxnFlow [10]. These models explicitly generate reaction pathways, allowing for a direct comparison of the synthesis costs of the proposed molecules. Table 1 reports the results of the evaluation on the sEH proxy task with a reward threshold of 8.0. Additional results for the GSK3$\beta$ and JNK3 tasks, as well as for the sEH task with a lower threshold of 7.2, are in Appendix F.

Introducing Synthesis Cost Guidance (C) into SCENT consistently reduces the synthesis cost across all settings while also improving the average reward. Importantly, it significantly increases the number of high-reward scaffolds in the MEDIUM and LARGE settings across all tasks. In the sEH task at the 8.0 threshold, Cost Guidance also substantially increases the number of high-reward modes; however, a slight decrease is observed at the 7.2 threshold. This indicates a shift toward more exploitative behavior, with the model focusing on high-reward regions. We analyze this trend in detail in Section 4.4, where we also motivate the introduction of the Exploitation Penalty.

Adding the Exploitation Penalty (P) to SCENT with Synthesis Cost Guidance (C+P) leads to consistent performance improvements across all settings and tasks, without sacrificing cost reduction. Notably, P recovers and surpasses the number of discovered modes at the 7.2 threshold, effectively mitigating the over-exploitation introduced by Synthesis Cost Guidance.

Incorporating the Dynamic Library (D) into SCENT with Synthesis Cost Guidance (C+D) yields further gains across all settings. These improvements are attributed to better credit assignment in compressed trajectories. The combination of P and D proves especially effective in the SMALL setting, suggesting complementary roles in improving exploration under constrained conditions.

Ultimately, SCENT with C+D+P in SMALL and C+P in MEDIUM and LARGE achieves the best overall performance in terms of diversity, quality, and cost of synthesis. These results highlight Cost Guidance's dual role in reducing the cost of high-reward molecules and enabling effective scaling to larger building block libraries, as explored in Section 4.5.

The qualitative analysis of how Synthesis Cost Guidance and Dynamic Library changes the generated trajectories can be found in Appendix G.

## 4.3   Synthesis Cost Reduction Mechanism

We analyze how Cost Guidance reduces the average cost of the reaction path generated by $P_F$. As shown in Figure 4-a, Synthesis Cost Guidance shortens trajectory lengths, thereby decreasing overall path costs. Furthermore, Figure 4-b and -d demonstrate that Cost Guidance selects cheaper fragments and significantly limits the use of more expensive ones. Finally, Figure 4-c illustrates that the Synthesis Cost Guidance relies on a subset of fragments throughout training, which correlates with the frequent use of cheaper molecules. The corresponding analysis for the SMALL setting is provided in Appendix H. We perform an ablation study on different cost reduction methods in Appendix J.2.2, showing that Synthesis Cost Guidance consistently identifies more modes and higher-reward scaffolds, at lower costs.

## 4.4   Synthesis Cost Guidance Exploitativeness Analysis

Table 1: Template-based GFlowNets on the sEH proxy across SMALL, MEDIUM, and LARGE settings. SCENT (C) significantly reduces molecule cost and boosts scaffold and mode discovery in MEDIUM and LARGE. Dynamic Library (C+D) improves performance in all settings, while adding Exploitation Penalty (+P) yields the best overall results, making SCENT the strongest method.

| | model | online discovery | | inference sampling | |
| --- | --- | --- | --- | --- | --- |
| | | modes $> 8.0 \uparrow$ | scaff. $> 8.0 \uparrow$ | reward $\uparrow$ | cost $> 8.0 \downarrow$ |
| SMALL | RGFN | 538 $\pm 21$ | 5862 $\pm 354$ | 7.34 $\pm 0.06$ | 37.7 $\pm 2.0$ |
| | SynFlowNet | 10.0 $\pm 0.8$ | 27.0 $\pm 8.6$ | 6.89 $\pm 0.29$ | - |
| | RxnFlow | 11.3 $\pm 5.0$ | 29.7 $\pm 12.7$ | 4.18 $\pm 0.72$ | - |
| | SCENT (w/o C) | 510 $\pm 26$ | 5413 $\pm 334$ | 7.38 $\pm 0.02$ | 37.1 $\pm 1.0$ |
| | SCENT (C) | 478 $\pm 11$ | 5150 $\pm 87$ | 7.43 $\pm 0.02$ | 23.7 $\pm 4.0$ |
| | SCENT (D) | 557 $\pm 6$ | 6496 $\pm 34$ | 7.42 $\pm 0.02$ | 32.8 $\pm 1.9$ |
| | SCENT (P) | 644 $\pm 6$ | 7840 $\pm 169$ | 7.33 $\pm 0.03$ | 36.0 $\pm 1.4$ |
| | SCENT (C+D) | 596 $\pm 19$ | 7148 $\pm 497$ | 7.56 $\pm 0.05$ | **19.7** $\pm \mathbf{2.0}$ |
| | SCENT (C+P) | 595 $\pm 20$ | 6839 $\pm 360$ | 7.42 $\pm 0.05$ | 20.7 $\pm 2.6$ |
| | SCENT (C+D+P) | **715** $\pm \mathbf{15}$ | **8768** $\pm \mathbf{363}$ | **7.66** $\pm \mathbf{0.03}$ | 20.7 $\pm 3.0$ |
| MEDIUM | RGFN | 4755 $\pm 541$ | 10638 $\pm 918$ | 7.08 $\pm 0.08$ | 1268 $\pm 71$ |
| | SynFlowNet | 288 $\pm 28$ | 299 $\pm 32$ | 6.38 $\pm 0.03$ | 1952 $\pm 548$ |
| | RxnFlow | 26.3 $\pm 3.9$ | 27.3 $\pm 2.5$ | 6.3 $\pm 0.02$ | - |
| | SCENT (w/o C) | 9310 $\pm 863$ | 11478 $\pm 823$ | 7.31 $\pm 0.09$ | 1463 $\pm 62$ |
| | SCENT (C) | 17705 $\pm 4224$ | 52340 $\pm 4303$ | 7.74 $\pm 0.04$ | 1163 $\pm 147$ |
| | SCENT (D) | 8629 $\pm 889$ | 10605 $\pm 1034$ | 7.21 $\pm 0.16$ | 1522 $\pm 139$ |
| | SCENT (P) | 13270 $\pm 647$ | 17093 $\pm 570$ | 7.32 $\pm 0.04$ | 1462 $\pm 16$ |
| | SCENT (C+D) | 17241 $\pm 679$ | 52607 $\pm 1352$ | 7.71 $\pm 0.03$ | 1141 $\pm 65$ |
| | SCENT (C+P) | **37714** $\pm \mathbf{3430}$ | **90068** $\pm \mathbf{9010}$ | 7.81 $\pm 0.04$ | 1230 $\pm 121$ |
| | SCENT (C+D+P) | 32761 $\pm 1483$ | 86081 $\pm 3171$ | 7.75 $\pm 0.08$ | **1117** $\pm \mathbf{33}$ |
| LARGE | RGFN | 5242 $\pm 123$ | 6215 $\pm 195$ | 7.09 $\pm 0.11$ | 1725 $\pm 153$ |
| | SynFlowNet | 278 $\pm 282$ | 385 $\pm 401$ | 6.39 $\pm 0.19$ | - |
| | RxnFlow | 24.5 $\pm 2.5$ | 25.0 $\pm 2.0$ | 6.14 $\pm 0.08$ | - |
| | SCENT (w/o C) | 7171 $\pm 291$ | 8767 $\pm 429$ | 7.13 $\pm 0.12$ | 1678 $\pm 63$ |
| | SCENT (C) | 12375 $\pm 264$ | 36930 $\pm 5455$ | 7.52 $\pm 0.09$ | 1267 $\pm 159$ |
| | SCENT (D) | 6384 $\pm 478$ | 7956 $\pm 690$ | 7.2 $\pm 0.11$ | 1656 $\pm 40$ |
| | SCENT (P) | 4594 $\pm 414$ | 8475 $\pm 737$ | 7.22 $\pm 0.05$ | 1779 $\pm 85$ |
| | SCENT (C+D) | 17138 $\pm 1341$ | 46443 $\pm 2080$ | 7.68 $\pm 0.11$ | **1256** $\pm \mathbf{107}$ |
| | SCENT (C+P) | **26367** $\pm \mathbf{3193}$ | 46202 $\pm 10242$ | 7.76 $\pm 0.03$ | 1291 $\pm 66$ |
| | SCENT (C+D+P) | 24434 $\pm 4223$ | **47816** $\pm \mathbf{5538}$ | **7.78** $\pm \mathbf{0.1}$ | 1399 $\pm 163$ |

The results in Section 4.2, together with the cost reduction mechanism detailed in Section 4.3, suggest that Synthesis Cost Guidance enhances the exploitative behavior of $P_F$. To validate this, we plot how often $P_F$ revisits previously discovered high-reward scaffolds during training in Figure 5. SCENT with Synthesis Cost Guidance (C) revisits the scaffolds drastically more often than SCENT without Synthesis Cost Guidance, confirming its increased exploitativeness. While this behavior promotes the discovery of high-reward scaffolds, it can reduce the diversity of discovered modes in certain settings, as observed in Appendix F. To address this trade-off, we introduced the Exploitation Penalty in Section 3.3, which drastically increases with the number of discovered modes and successfully limits scaffolds revisits in Figure 5.

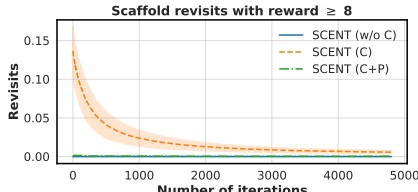

Figure 5: Revisit frequency of high-reward scaffolds increases sharply with Synthesis Cost Guidance (C), indicating greater exploitative behavior. Introducing the Exploitation Penalty (P) effectively reduces this revisit ratio.

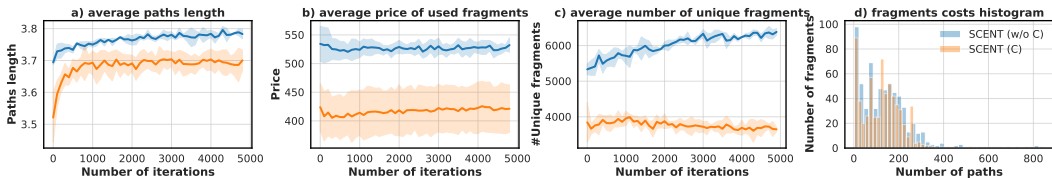

Figure 4: Synthesis Cost Guidance reduces trajectory length (a), fragments cost (b), and reliance on expensive fragments (d), while concentrating on a smaller fragment subset (c). Results are smoothed over the last 100 iterations.

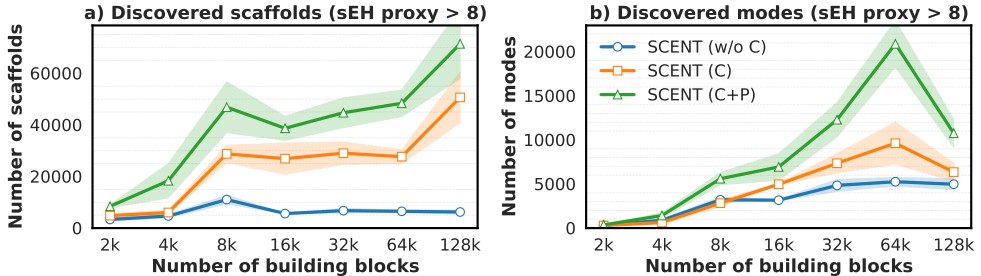

Figure 6: Number of discovered a) scaffolds and b) modes as a function of the size of building block library $M$. The numbers are reported for the 3000th training iteration.

### 4.5 Synthesis Cost Guidance Helps in Scaling

We conducted additional experiments to assess how the size of the building block library $M$ affects the performance gap between SCENT, SCENT without Cost Guidance, and SCENT with the Exploitation Penalty. As shown in Figure 6, Synthesis Cost Guidance consistently increases the number of discovered scaffolds across all library sizes (2k–128k). For mode discovery, the gap widens up to $M = 64k$, then narrows at 128k.

Although initially counterintuitive that Cost Guidance improves mode-seeking, Section 4.3 suggests that it enables more efficient navigation of fragment space, reducing the number of fragments used. Previous work [9, 8] shows that reducing library size via random cropping can help, but the effect is modest compared to our gains. Similarly, removing high-cost fragments improves diversity only marginally (Appendix J.2.2). These findings support our conclusion that Cost Guidance learns to prioritize fragments that are both cost-efficient and reward-relevant.

### 4.6 Synthesizability-Oriented Metrics

We compare SCENT with leading GFlowNet baselines for molecular generation: RGFN and the fragment-based GFlowNet (FGFN) from Bengio et al. [7]. We also evaluate FGFN (SA), trained to jointly optimize the sEH proxy and the Synthetic Accessibility (SA) score. As shown in Figure 7, template-based models substantially improve synthesizability. This is most evident in the AiZynthFinder success rate [48], where FGFN fails completely and FGFN (SA) performs poorly relative to RGFN and SCENT. Notably, although FGFN (SA) achieves the lowest average SA score, it still struggles on AiZynthFinder, underscoring the limitations of heuristic synthesizability metrics. SCENT achieves the highest average reward and outperforms RGFN in synthesizability, which we attribute to its preference for shorter reaction pathways (Section 4.3), resulting in reduced molecular weight that correlates with the synthesibility metrics [49].

## 5 Conclusion

In this paper, we presented SCENT, a template-based approach for molecular generation within the GFlowNet framework. SCENT introduces three new mechanisms that collectively enhance exploration of chemical space: (1) Resursive Cost Guidance, a general strategy for guiding GFlowNet's backward policy, which we utilized to improve computational efficiency by removing costly par-

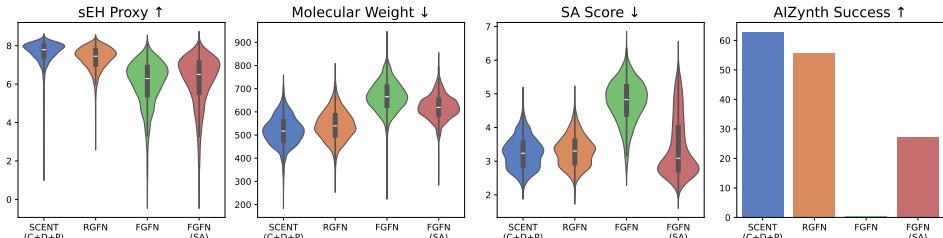

Figure 7: Synthesibility metrics for GFlowNet models. SCENT and RGFN were trained on SMALL.

ent state computation, and to focus sampling on molecules with low synthesis cost, (2) Dynamic Library building blocks, which improve exploration efficiency—especially with smaller fragment libraries—and enable the generation of non-linear synthesis trees, and (3) an Exploitation Penalty, which avoids repeated visits to previously explored candidates. Through empirical evaluation, we demonstrated that combining these strategies leads to state-of-the-art results in template-based molecular generation across diverse experimental settings, from small-scale fragment libraries suitable for rapid, automated synthesis to large-scale libraries that create vast search spaces.

# 6 Limitations

While our study demonstrates the effectiveness of SCENT in generating cost-efficient and synthesizable molecules, several limitations remain.

First, the synthesis cost estimates are simplified and heuristic, omitting factors such as solvent usage, purification steps, hazardous reagents, reaction conditions, and scalability. Consequently, the current cost model should be regarded as a first-order approximation. More accurate estimation would require lab-specific calibration, including robust yield prediction models and vendor-specific building block pricing.

SCENT's performance varies across different biological targets, indicating that its effectiveness may depend on task-specific factors. Identifying and addressing these bottlenecks is an important direction for future work. Key areas of improvement include refining the exploitation behavior of the Dynamic Library, and enhancing the generalizability of the cost predictor model used in Synthesis Cost Guidance. Additionally, we do not assess performance as the number of reaction templates increases, an important scalability consideration, as this expansion significantly enlarges the action space and introduces additional learning challenges.

Our framework emphasizes synthesizability and cost, but does not incorporate toxicity or broader ADMET properties, which are critical for downstream drug development. Finally, we acknowledge the absence of wet-lab validation, which remains the definitive benchmark for evaluating the practical utility of our approach. We leave this to future work.

# Acknowledgements

The research of P. Gaiński was supported by the National Science Centre (Poland), grant no. 2022/45/B/ST6/01117. O. Boussif was supported by the National Research Council Canada (NRC) AI4D program. We gratefully acknowledge Polish high-performance computing infrastructure PLGrid (HPC Center: ACK Cyfronet AGH) for providing computer facilities and support within computational grant no. PLG/2024/017716. The research was also enabled by computational resources provided by the Digital Research Alliance of Canada (`https://alliancecan.ca`) and Mila (`https://mila.quebec`). We thank Marcin Sendera and Moksh Jain for their support during the early stages of developing this paper.

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

# A  SCENT Details

## A.1  Pseudocode

---

**Algorithm 1** SCENT training

---

**Input:** building block library $M$, reactant patterns $R$, forward policy $P_F$, flags: $C$, $P$, and $D$.
**Initialize:** sampling policy $P'_F$, backward policy $P_B$.
   **if** $C$ **then**: $P_B = P_B^D \times P_B^S$, **else**: $P_B = P_B^D$.
   **if** $P$ **then**: $P'_F = P_F \times P_E$, **else**: $P'_F = P_F$ with $\epsilon$-greedy.
  **for** $i = 1$ **to** $N$ **do**
   Sample forward trajectories $T_F$ from $P'_F$.
   Sample replay trajectories $T_B$ from prioritized replay buffer using $P_B$.
   Update $P_F$ to minimize $\mathcal{L}_{TB}$ over $T = T_F \cup T_B$.
   Update Decomposability Guidance model $\hat{c}_B^D$.
   **if** $C$ **then**: Update Synthesis Cost Guidance model $\hat{c}_B^S$, **else**: do nothing.
   **if** $P$ **then**: Update Exploitation Penalty counter $n(s, a)$, **else**: do nothing.
   **if** $D$ **then**: Update Dynamic Library utility $U$ and every $T$ steps update $M$, **else**: do nothing.
  **end for**

---

## A.2  Forward Policy

The backbone of SCENT's $P_F$ is an adapted graph transformer model $f$ from [50]. It takes as input a partially constructed molecule $m$ (represented as a graph) and an optional reaction template $r$ (represented as a one-hot vector), and outputs an embedding $f(m, r) \in \mathbb{R}^D$. The building blocks from $M$ are embedded using a linear combination of the ECFP4 fingerprint $e(m)$ and a learnable embedding $v(m)$ uniquely associated with a given building block: $h(m) = We(m) + v(m)$.

A molecule (and a corresponding synthesis pathway) is composed in a sequence of stages that are further described in this section. All stages linearly transform the state embeddings obtained with a backbone model $f$ using weight matrices $W_i$.

**Select an initial building block.** In SCENT, the trajectory starts with a fragment $m$ sampled from $M$ with probability:

$$p(m_i) = \sigma_i^{|M|}(\mathbf{l}), \; l_i = \phi(W_1 f(\emptyset, \emptyset))^T h(m_i), \tag{14}$$

where $\phi$ is GELU activation function [51] and $\sigma^k$ is a standard softmax over the logits vector $\mathbf{l} \in \mathbb{R}^k$ of length $k$:

$$\sigma_i^k(\mathbf{l}) = \frac{\exp(l_i)}{\sum_{j=1}^k \exp(l_j)}.$$

**Select the reaction template.** The next step is to sample a reaction $r_i \in R$ that can be applied to the molecule $m$:

$$p(r_i|m) = \sigma_i^{|R|+1}(\mathbf{l}), \; l_i = \phi(W_2 f(m, \emptyset))^T v(r_i), \tag{15}$$

where $v(r_i)$ is a learnable embedding for the $i$-th reaction. Choosing the stop action with the index $|R| + 1$ ends the trajectory.

**Select a set of reactants** We want to find a building block $m_i \in M$ that will react with $m$ and reactants $M'$ by the reaction template $r$. The probability of selecting $m_i$ is:

$$p(m_i|m, r, M') = \sigma_i^{|M|}(\mathbf{l}),$$
$$l_i = \phi(W_3 g(m, r, M'))^T h(m_i)$$
$$g(m, r, M') = f(m, r) + \sum_{m'_i \in M'} (h(m'_i) + v(i)),$$

where $v(i)$ is a learnable positional embedding for the $i$-th position in the reaction template. If the reaction $r$ does not require any additional reactant, this step is skipped.

**Perform the reaction and select one of the resulting molecules.** In this step, we apply the reaction $r$ to a set of chosen fragments (it can be empty). We then sample the outcome molecule $m$ from a set of possible outcomes $O$:

$$p(m_i) = \sigma_i^{|O|}(\mathbf{1}), \ l_i = \phi(W_4(f(m_i, \emptyset)))^T w, \tag{16}$$

where $w \in \mathbb{R}^D$ is a learnable parameter.

### A.3  Cost Prediction Model Training

We use two cost prediction models in SCENT: synthesis cost prediction model $\hat{c}_B^S$ and decomposability prediction model $\hat{c}_B^D$. The size of mini-batches, the dataset size $N$, and the negative sampling ratio were manually chosen to minimize the cost prediction validation losses gathered during the training.

#### A.3.1  Synthesis Cost Model

The cost prediction model $\hat{c}_B^S$ is a simple multi-layer perception on top of ECFP4 fingerprint $e(s) \in \mathbb{R}^{2048}$:

$$\hat{c}(s) = W_1\phi(W_2 e(s)), \ W_1 \in \mathbb{R}^{1 \times 128}, \ W_2 \in \mathbb{R}^{128 \times 2048}, \tag{17}$$

where $\phi$ is the GELU activation function [51].

We train the model $\hat{c}_B^S$ using the procedure described in Section 3.2. Given an updated dataset with synthesis cost estimates $D$, we perform five mini-batch updates of $\hat{c}_B^S$ using mini-batches of size 1024, the learning rate of 0.01 and the mean square error loss $\mathcal{L}_{cost} = \sum_{c_s \in D'} ||c_s - \hat{c}_B^S||_2^2$, where $D'$ is a mini-batch of $D$. To stabilize training, we standardize all costs in $D$ using the mean and variance of costs from the building block library $M$.

#### A.3.2  Decomposability Model

The decomposability model $\hat{c}_B^D$ is very similar to the synthesis cost model, except that it encodes the current number of reactions used in the molecule $s$. The "cost" in the context of decomposability is 0 if the given molecule can be decomposed into building blocks from $M$ (within some number of steps) and 1 otherwise. In practice, we train the model $\hat{c}_B^D$ to predict the decomposability label $\in \{0, 1\}$ using binary cross entropy $\mathcal{L}_{cost}$. The labels are gathered similarly to the synthesis cost values, using the trajectories encountered in the GFlowNet forward sampling. We update $\hat{d}(s)$ with five minibatches of size 1024 and a learning rate of 0.005. In addition, we ensure that at least 20% of the samples of molecules for training $\hat{d}(s)$ are not decomposable.

## B  Training Details

All the models sampled 320,000 forward trajectories during the training in total in SMALL and MEDIUM settings and 256,000 in LARGE. All forward policies were trained using the Trajectory Balance objective [31] and their parameters were optimized using Adam [52]. All methods were trained using their built-in action embedding mechanisms and on a maximum number of reactions equal to 4. Hyperparameters were chosen semi-manually or using small grid searches to maximize the number of high-reward modes in the SMALL setting in the sEH proxy.

### B.1  SCENT

We based the SCENT implementation on the official RGFN codebase (`https://github.com/koziarskilab/RGFN`). We followed most of the hyperparameter choices from the official implementation of RGFN [8]. We set the number of sampled forward trajectories to 64, and the number of trajectories sampled from the prioritized replay buffer to 32. We set the $\beta$ to 8 for sEH, and 48 for GSK3$\beta$ and JNK3. We used uniform $\epsilon$-greedy exploration with $\epsilon = 0.05$ for all our experiments, except for those with other exploration techniques (e.g. Exploitation Penalty).

All SCENT instances were trained using Decomposability Guidance, either solely or with Synthesis Cost Guidance using Equation (11).

**Decomposability Guidance**   To predict the decomposability of the molecule, we use the model described in Appendix A.3. We set the cost temperature (Equation (10)) to $\alpha_D = 5$.

**Synthesis Cost Guidance**   The model for Synthesis Cost Guidance is described in Appendix A.3. We set the cost temperature to $\alpha = 5$.

**Dynamic Building Block Library**   We update the dynamic building block library every $T = 1000$ iterations for a maximum of $N_{\text{add}} = 10$ additions (that is, we do this until the end of training). The number of molecules added to the library every time it is updated is $L = 400$.

**Exploitation Penalty**   We set $\epsilon = 3$ in Equation (12) and schedule $\gamma$ in two ways: 1) so that it linearly decays to zero after $N$ iterations, and 2) it grows with the trajectory length (each step increases it by some constant factor $\Delta\gamma$). We set $N = \infty$ for SCENT that uses Dynamic Library and $N = 1000$ for the rest of the runs. The growing delta $\Delta\gamma = 0.2$, while the initial $s_0$ temperature $\gamma_0 = 1.0$. Note that for JNK3 in the MEDIUM setting, we used $4500$ iterations for exploitation penalty.

## B.2   RGFN

We use the official implementation of RGFN [8] with mostly default parameters. We changed the number of sampled forward trajectories to $64$, and the number of trajectories sampled from the prioritized replay buffer to $32$. We set the $\beta$ to $8$ for sEH, and $48$ for GSK3$\beta$ and JNK3.

## B.3   SynFlowNet

We followed the setting from the paper [9] and the official repository. We used **MaxLikelihood** or REINFORCE backward policy with **disabled** or enabled replay buffer. We grid searched GFlowNet reward temperature $\beta$: $\{\mathbf{32}, 64\}$ for sEH and $\beta \in \{\mathbf{16}, 32\}$ for GSK3$\beta$ and JNK3.

We trained SynFlowNet using $32$ forward trajectories and $8000$ iterations on LARGE setting, and $64$ forward trajectories and $5000$ iterations for SMALL and MEDIUM.

## B.4   RxnFlow

We used the default parameters from the official implementation and adapted 1) the `action_subsampling_ratio` so that the number of sampled actions is close to the number used in their paper, and 2) GFlowNet $\beta$ temperatures. We set `action_subsampling_ratio=1.0` for the SMALL setting, $0.2$ for MEDIUM, and $0.1$ for LARGE. We set the $\beta$ sampling distribution to Uniform$(16, 64)$ for sEH, and Uniform$(16, 48)$ for GSK3$\beta$ and JNK3.

# C   Hyperparameters Sensitivity

To assess SCENT (C+D+P) sensitivity to hyperparameters, we performed an ablation study on sEH MEDIUM, varying each hyperparameter around the chosen values (center in tables) over 3k iterations. Results in Table 2 indicate SCENT is generally robust, with no catastrophic performance drops. However, some hyperparameters notably affect outcomes; for example, simple tuning of $\alpha_D$ can boost discovered modes by  20%. This highlights potential for further improvements through targeted tuning, left for future work.

# D   Fragment analysis

In this section, we showcase characteristics of the fragments we used in this paper for all settings. We focus on molecular weight, lipophilicity (logP), the number of Hydrogen bond donors (HBD) and acceptors (HBA), the Topological polar surface area (TPSA) and the number of rotatable bonds (See Figure 8).

Table 2: Sensitivity of SCENT (C+D+P) to hyperparameters. Each hyperparameter was varied around the chosen values (center in tables). The results are reported for MEDIUM settings after 3k training iterations.

| model | online discovery | | inference sampling | |
|---|---|---|---|---|
| | modes $> 8.0 \uparrow$ | scaff. $> 8.0 \uparrow$ | reward $\uparrow$ | cost $> 8.0 \downarrow$ |
| Synthesis Cost Guidance (temperature $\alpha_S$) | | | | |
| $\alpha_S = 2$ | $16036 \pm 1153$ | $22292 \pm 1652$ | $7.55 \pm 0.05$ | $1300 \pm 82$ |
| $\alpha_S = 5$ | $16373 \pm 1827$ | $39721 \pm 3574$ | $7.61 \pm 0.03$ | $1163 \pm 81$ |
| $\alpha_S = 10$ | $13566 \pm 4207$ | $73916 \pm 2516$ | $7.87 \pm 0.04$ | $898 \pm 42$ |
| Decomposability Guidance (temperature $\alpha_D$) | | | | |
| $\alpha_D = 2$ | $20023 \pm 1484$ | $45483 \pm 2490$ | $7.74 \pm 0.04$ | $1137 \pm 130$ |
| $\alpha_D = 5$ | $16373 \pm 1827$ | $39721 \pm 3574$ | $7.61 \pm 0.03$ | $1163 \pm 81$ |
| $\alpha_D = 10$ | $20444 \pm 2055$ | $44938 \pm 3111$ | $7.74 \pm 0.08$ | $1141 \pm 80$ |
| Dynamic Library (number of added molecules $L$) | | | | |
| $L = 200$ | $19700 \pm 1919$ | $49119 \pm 6425$ | $7.72 \pm 0.05$ | $1087 \pm 183$ |
| $L = 400$ | $16373 \pm 1827$ | $39721 \pm 3574$ | $7.61 \pm 0.03$ | $1163 \pm 81$ |
| $L = 800$ | $18373 \pm 987$ | $48016 \pm 4302$ | $7.71 \pm 0.08$ | $1060 \pm 66$ |
| Dynamic Library (frequency of updates $T$) | | | | |
| $T = 500$ | $15670 \pm 2482$ | $39701 \pm 3676$ | $7.66 \pm 0.12$ | $1130 \pm 106$ |
| $T = 1000$ | $16373 \pm 1827$ | $39721 \pm 3574$ | $7.61 \pm 0.03$ | $1163 \pm 81$ |
| $T = 2000$ | $16148 \pm 1753$ | $39099 \pm 3341$ | $7.66 \pm 0.08$ | $1141 \pm 82$ |
| Exploitation Penalty (initial $\gamma_0$) | | | | |
| $\gamma_0 = 0.5$ | $20690 \pm 1512$ | $47220 \pm 3676$ | $7.69 \pm 0.06$ | $1154 \pm 48$ |
| $\gamma_0 = 1.0$ | $16373 \pm 1827$ | $39721 \pm 3574$ | $7.61 \pm 0.03$ | $1163 \pm 81$ |
| $\gamma_0 = 2.0$ | $20655 \pm 1514$ | $45990 \pm 1593$ | $7.75 \pm 0.04$ | $1111 \pm 77$ |
| Exploitation Penalty ($\Delta\gamma$) | | | | |
| $\Delta\gamma = 0.0$ | $20025 \pm 762$ | $43432 \pm 796$ | $7.75 \pm 0.02$ | $1222 \pm 65$ |
| $\Delta\gamma = 0.2$ | $16373 \pm 1827$ | $39721 \pm 3574$ | $7.61 \pm 0.03$ | $1163 \pm 81$ |
| $\Delta\gamma = 0.4$ | $17189 \pm 2234$ | $42718 \pm 2799$ | $7.69 \pm 0.04$ | $1108 \pm 165$ |

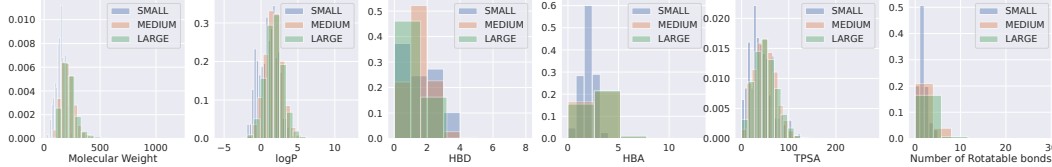

Figure 8: Exploratory analysis of chemical properties of fragments in the SMALL, MEDIUM and LARGE settings.

# E    Reaction Yields and Fragments Costs Estimates

**SMALL setting.** The reaction set for the SMALL setting is the extended reaction set from Koziarski et al. [8]. We added Bishler-Napieralski and Pictet-Spengler reactions, along with various aryl functionalizations, benzoxazole, benzimidazole, and benzothiazole formation reactions, Hantzsch thiazole synthesis, alkylation of aromatic nitrogen, and Williamson ether synthesis.

Building block prices for SMALL were obtained from the vendors online. If multiple vendors offered the same building block, only the cost of the cheapest option was considered. For compounds available in varying amounts, both the price per gram and the alternative size (e.g., 5, 10, 25 grams) were considered, and the smallest price per mmol was used. Compounds that were exclusively accessible

in other forms, such as salts, related functional groups, or containing a protected functional group were used with the SMILES fitted to the corresponding reactions. The cost of a product includes the total cost of the building blocks used, while additional expenses such as solvents, catalysts, and reagents are not accounted for. For the MEDIUM and LARGE settings, stock prices were obtained by automatically scraping Enamine's publicly available catalog.

For reactions where sufficient in-house experimental data was available (amide coupling, nucleophilic additions to isocyanates, Suzuki reaction, Buchwald–Hartwig coupling, Sonogashira coupling, and azide–alkyne cycloaddition), yields were calculated as the average of all recorded experimental yields corresponding to each SMARTS-based reaction template. For the remaining reactions in the dataset, yield estimates were derived from the average of reported literature yields for reactions matching the respective SMARTS templates.

**MEDIUM and LARGE setting.** We used Enamine building blocks subsets of size 64,000 for MEDIUM and 128,000 for LARGE settings. The Enamine REAL database was first filtered to retain only molecules with currently available cost information. Fragment selection was then performed randomly for LARGE setting, without any specific criteria, to ensure broad coverage of molecular space. The building blocks for MEDIUM setting were subsampled from LARGE one.

# F  Extended Comparison to Template-Based GFlowNets

## F.1  Results on sEH with Reward Threshold = 7.2

Despite tuning the SynFlowNet [9] parameters (Appendix B), we could not obtain satisfying results. Importantly, our experimental setting differs from the one used in their paper in the following way: 1) we use up to 4 reactions, they use 2 or 3, 2) they use a 10k subset of Enamine, our MEDIUM setting contains 64k molecules, 3) we set the reward thresholds higher (for sEH we set 8.0 instead of 7.2), and 4) we train the model for fewer trajectories. In Table 3, we report the results for sEH using a threshold from the SynFlowNet paper. The number of modes obtained seems consistent with those reported in Cretu et al. [9].

## F.2  Results on GSK3$\beta$

We benchmark SCENT on the GSK3$\beta$ proxy and observe that our approach consistently discovers more modes and high-reward scaffolds than alternative reaction-based GFlowNets (see Table 4). The exploitation penalty improves performance across all settings, while using cost biasing alone (SCENT (C)) is slightly outperformed by SCENT (w/o C), which does not use cost biasing. However, it is worth noting that SCENT (C) finds cheaper molecules, with the fully enabled SCENT (C+P) identifying the cheapest ones. In the largest setting, cost biasing increases the number of modes found, and adding the exploitation penalty further boosts performance.

## F.3  Results on JNK3

In Table 5 RGFN outperforms all SCENT variants in the smallest setting, consistently finding more diverse modes and high-reward scaffolds. While SCENT lags behind in terms of mode discovery, it compensates in terms of cost, since SCENT (C) samples the cheapest molecules. In the MEDIUM setting, SCENT (w/o C) outperforms the baseline methods and the fully enabled SCENT variants in terms of mode discovery, but lags behind in the number of high-reward scaffolds. That can be attributed to the fact that SCENT (C) samples are less diverse. However, SCENT (C+P) samples the cheapest molecules. Finally, in the largest setting, the fully enabled SCENT (C+P) finds the most number of modes and high-reward scaffolds, whereas SCENT (C) takes the spotlight in terms of cost, generating the cheapest molecules.

## F.4  Comparison of computational resources

In Table 6 report GPU runtimes (in minutes) for all template-based baselines on the sEH proxy, averaged over 3 random seeds using a V100 32GB GPU. All methods used a batch size of 64, except SynFlowNet, which required batch size 32 on the LARGE setting even after optimizing its policy code to fit within memory limits.

Table 3: Evaluation of different template-based approaches using sEH proxy in SMALL, MEDIUM and LARGE settings, but using lower reward threshold ($> 7.2$) equivalent to the one used in SynFlowNet paper [9].

| | model | online discovery | | inference sampling | |
|---|---|---|---|---|---|
| | | modes $> 7.2 \uparrow$ | scaff. $> 7.2 \uparrow$ | reward $\uparrow$ | cost $> 7.2 \downarrow$ |
| **SMALL** | RGFN | 11320 ±168 | 76742 ±693 | 7.34 ±0.06 | 35.3 ±1.6 |
| | SynFlowNet | 559 ±217 | 4795 ±2477 | 6.89 ±0.29 | 22.6 ±7.3 |
| | RxnFlow | 472 ±259 | 734 ±270 | 4.18 ±0.72 | 52.3 ±5.9 |
| | SCENT (w/o C) | 13862 ±422 | 114522 ±1558 | 7.38 ±0.02 | 38.8 ±1.0 |
| | SCENT (C) | 11978 ±197 | 108262 ±1355 | 7.43 ±0.02 | 23.5 ±2.8 |
| | SCENT (C+D) | 11877 ±48 | 110522 ±1632 | 7.56 ±0.05 | **18.9 ±2.8** |
| | SCENT (C+P) | 14399 ±1149 | **137443 ±1850** | 7.42 ±0.05 | 19.5 ±1.3 |
| | SCENT (C+D+P) | **14631 ±206** | 137426 ±1055 | **7.66 ±0.03** | 20.9 ±1.7 |
| **MEDIUM** | RGFN | 45937 ±3108 | 77876 ±3078 | 7.08 ±0.08 | 1386 ±64 |
| | SynFlowNet | 32310 ±682 | 35481 ±643 | 6.38 ±0.03 | 1666 ±109 |
| | RxnFlow | 2458 ±46 | 2613 ±45 | 6.3 ±0.02 | 1645 ±56 |
| | SCENT (w/o C) | 126409 ±4001 | 141284 ±2833 | 7.31 ±0.09 | 1594 ±59 |
| | SCENT (C) | 108921 ±17019 | 212402 ±1361 | 7.74 ±0.04 | 1151 ±140 |
| | SCENT (C+D) | 104702 ±3536 | 213750 ±1818 | 7.71 ±0.03 | 1150 ±60 |
| | SCENT (C+P) | **158578 ±3617** | **277811 ±4383** | **7.81 ±0.04** | 1230 ±132 |
| | SCENT (C+D+P) | 144224 ±2329 | 274380 ±1388 | 7.75 ±0.08 | **1096 ±29** |
| **LARGE** | RGFN | 54255 ±1573 | 59210 ±1146 | 7.09 ±0.11 | 1766 ±50 |
| | SynFlowNet | 20186 ±13305 | 36656 ±22657 | 6.39 ±0.19 | 1545 ±70 |
| | RxnFlow | 1615 ±60 | 1708 ±67 | 6.14 ±0.08 | 1800 ±22 |
| | SCENT (w/o C) | 95457 ±1292 | 104789 ±1684 | 7.13 ±0.12 | 1776 ±50 |
| | SCENT (C) | 83729 ±1157 | 157331 ±1783 | 7.52 ±0.09 | **1230 ±159** |
| | SCENT (C+D) | 90133 ±3566 | 164962 ±2931 | 7.68 ±0.11 | 1267 ±116 |
| | SCENT (C+P) | **140654 ±6121** | 198503 ±5047 | 7.76 ±0.03 | 1250 ±61 |
| | SCENT (C+D+P) | 136677 ±3734 | **199216 ±5986** | **7.78 ±0.1** | 1362 ±146 |

RGFN shares a part of the codebase with SCENT, allowing for direct comparison. We observe consistent improvements across all settings, particularly in the SMALL configuration. While SynFlowNet is faster on SMALL and MEDIUM, it struggles on LARGE, even with our memory optimizations. However, differences in codebases make direct comparisons less definitive.

Note that our implementation is not optimized for runtime and could benefit from further improvements.

## G   Qualitative Pathways Analysis

To visualize the Synthesis Cost Guidance in practice, we gathered training trajectories from SCENT (C) and SCENT (w/o C) that lead to the same high-rewarded molecule in the SMALL setting.

We observed that, on average, cost-guided trajectories are  10% cheaper. We selected a few example trajectory pairs and noticed that the cost reduction usually occurs by changing the order of the reactions: introducing expensive fragments later in the synthesis path, which decreases the product losses due to imperfect yields, and as a result increases the cost-efficiency. In Appendix G, SCENT (C) chooses the most expensive building block in the sequence ($12.56) at the beginning of the synthesis, while SCENT (C) utilizes it in the second step, effectively reducing the synthesis cost by almost 28%. Cost-guided model also occasionally prefers cheaper fragments, e.g., "Brc1c[nH]cn1" instead of "Ic1c[nH]cn1". It is worth emphasizing that the fragments in the SMALL setting are already low cost, which is the reason why most gains are observed due to reaction order swapping.

Similarly, we compared the Dynamic Library, SCENT (C+D), to SCENT (C), and observed that on average the trajectories are 12% shorter and 6% cheaper. We selected few examples that visualized the compressed trajectories. We have found that the Dynamic Library enables the generation of

Table 4: Evaluation of different template-based approaches using GSK3$\beta$ proxy in SMALL, MEDIUM and LARGE settings. In all metrics, except "reward", we consider molecules with reward $> 0.8$. We observe that SCENT (C) that uses Cost Guidance can greatly reduce the cost of the sampled molecules, while drastically increasing the number of scaffolds and modes discovered during the training (in MEDIUM and LARGE settings). Dynamic Library (C+D) significantly improves the performance of the model in the SMALL setting. Using exploitation penalty (+P) further boosts the results, making SCENT by far the most powerful evaluated method in all settings.

| | model | online discovery | | inference sampling | |
|---|---|---|---|---|---|
| | | modes $> 0.8 \uparrow$ | scaff. $> 0.8 \uparrow$ | reward $\uparrow$ | cost $> 0.8 \downarrow$ |
| SMALL | RGFN | 285 $\pm 14$ | 1870 $\pm 106$ | 0.7 $\pm 0.01$ | 42.3 $\pm 2.7$ |
| | SynFlowNet | 0.0 $\pm 0.0$ | 0.0 $\pm 0.0$ | 0.57 $\pm 0.01$ | - |
| | RxnFlow | 2.0 $\pm 0.0$ | 2.0 $\pm 0.0$ | 0.51 $\pm 0.05$ | - |
| | SCENT (w/o C) | 290 $\pm 17$ | 1911 $\pm 84$ | 0.7 $\pm 0.01$ | 44.1 $\pm 5.1$ |
| | SCENT (C) | 287 $\pm 9$ | 1856 $\pm 147$ | 0.7 $\pm 0.0$ | 34.9 $\pm 4.5$ |
| | SCENT (D) | 319 $\pm 8$ | 2196 $\pm 30$ | 0.71 $\pm 0.01$ | 38.0 $\pm 1.9$ |
| | SCENT (P) | 330 $\pm 6$ | **2447** $\pm$**94** | 0.7 $\pm 0.01$ | 45.8 $\pm 1.8$ |
| | SCENT (C+D) | 306 $\pm 14$ | 2138 $\pm 25$ | **0.73** $\pm$**0.01** | 30.3 $\pm 1.3$ |
| | SCENT (C+P) | 294 $\pm 18$ | 2000 $\pm 109$ | 0.71 $\pm 0.0$ | **30.0** $\pm$**0.4** |
| | SCENT (C+D+P) | **336** $\pm$**6** | 2172 $\pm 92$ | 0.72 $\pm 0.01$ | 32.1 $\pm 3.1$ |
| MEDIUM | RGFN | 4037 $\pm 1278$ | 5225 $\pm 1122$ | 0.64 $\pm 0.01$ | 1251 $\pm 211$ |
| | SynFlowNet | 681 $\pm 184$ | 1015 $\pm 393$ | 0.58 $\pm 0.02$ | 1565 $\pm 415$ |
| | RxnFlow | 64.0 $\pm 14.8$ | 88.0 $\pm 11.2$ | 0.58 $\pm 0.0$ | 1602 $\pm 59$ |
| | SCENT (w/o C) | 6430 $\pm 2427$ | 10010 $\pm 2537$ | 0.66 $\pm 0.03$ | 1177 $\pm 187$ |
| | SCENT (C) | 5945 $\pm 4009$ | 50435 $\pm 13360$ | 0.74 $\pm 0.03$ | 917 $\pm 200$ |
| | SCENT (D) | 5668 $\pm 2149$ | 8548 $\pm 2297$ | 0.64 $\pm 0.02$ | 1217 $\pm 90$ |
| | SCENT (P) | 6494 $\pm 1812$ | 11493 $\pm 1929$ | 0.66 $\pm 0.02$ | 1164 $\pm 39$ |
| | SCENT (C+D) | 9430 $\pm 4284$ | 64721 $\pm 15868$ | 0.77 $\pm 0.01$ | 776 $\pm 247$ |
| | SCENT (C+P) | 8868 $\pm 1883$ | 92813 $\pm 34174$ | 0.77 $\pm 0.05$ | 815 $\pm 56$ |
| | SCENT (C+D+P) | **11857** $\pm$**4171** | **93761** $\pm$**39462** | **0.82** $\pm$**0.04** | **730** $\pm$**184** |
| LARGE | RGFN | 1748 $\pm 660$ | 2056 $\pm 638$ | 0.62 $\pm 0.02$ | 1370 $\pm 176$ |
| | SynFlowNet | 402 $\pm 237$ | 970 $\pm 687$ | 0.53 $\pm 0.01$ | - |
| | RxnFlow | 31.7 $\pm 5.6$ | 37.7 $\pm 5.0$ | 0.55 $\pm 0.0$ | - |
| | SCENT (w/o C) | 3366 $\pm 3503$ | 4873 $\pm 4494$ | 0.6 $\pm 0.07$ | 1842 $\pm 606$ |
| | SCENT (C) | 4295 $\pm 1468$ | 28867 $\pm 2418$ | 0.7 $\pm 0.01$ | 973 $\pm 119$ |
| | SCENT (D) | 2256 $\pm 472$ | 3185 $\pm 514$ | 0.57 $\pm 0.03$ | 1390 $\pm 239$ |
| | SCENT (P) | 1855 $\pm 234$ | 2604 $\pm 314$ | 0.6 $\pm 0.0$ | 1376 $\pm 68$ |
| | SCENT (C+D) | **8238** $\pm$**2716** | **37957** $\pm$**3277** | **0.74** $\pm$**0.01** | **693** $\pm$**107** |
| | SCENT (C+P) | 7890 $\pm 1869$ | 26553 $\pm 2121$ | 0.73 $\pm 0.01$ | 931 $\pm 197$ |
| | SCENT (C+D+P) | 5574 $\pm 708$ | 34595 $\pm 3254$ | 0.73 $\pm 0.03$ | 990 $\pm 151$ |

convergent synthetic pathways (Appendix G), which are responsible for the reduction of the synthesis cost. The application of the convergent synthetic strategy decreases the longest linear sequence (LLS) of steps, thereby increasing the overall yield.

In the MEDIUM settings, the SCENT variations diverged early on, so the set of shared molecules is very limited. Therefore, we matched high-rewarded molecules from SCENT (C) to closest high-rewarded molecule from SCENT (w/o C) with Tanimoto similarity $> 0.6$. We observed that, on average, the cost-guided trajectories are 20% cheaper. Thorough investigation of few selected trajectories pairs revealed that, differently from SMALL setting, in MEDIUM setting, the cost reduction comes mostly from the selection of cheaper fragments (Appendix G).

## H  Extended Synthesis Cost Guidance Analysis

In this section, we extend the analysis from Section 4.3 with results in the SMALL setting. Figure 12 shows that similarly to the MEDIUM setting, Synthesis Cost Guidance reduces the average length of

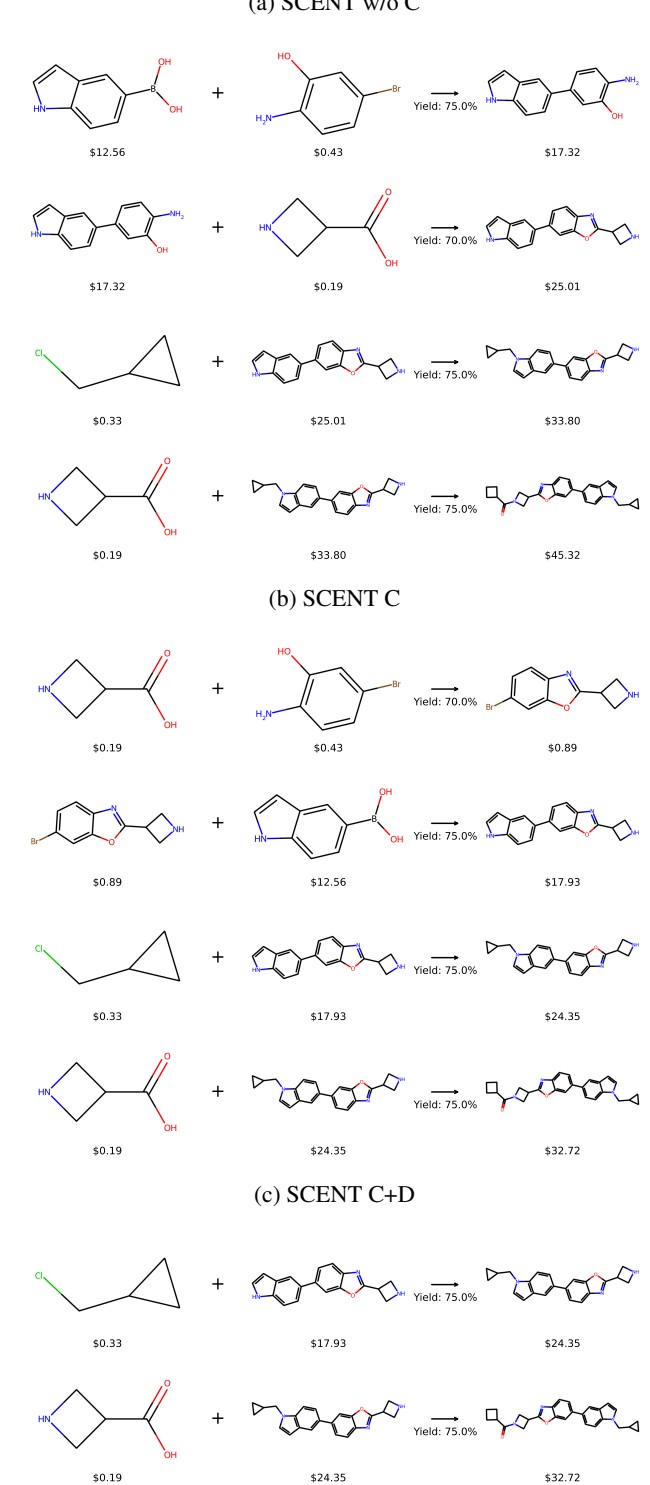

Figure 9: Three example pathways from SMALL setting leading to the same molecule. SCENT (C) is cheaper than SCENT (w/o C) by introducing the costly fragment ($12.56) later in the pathway.

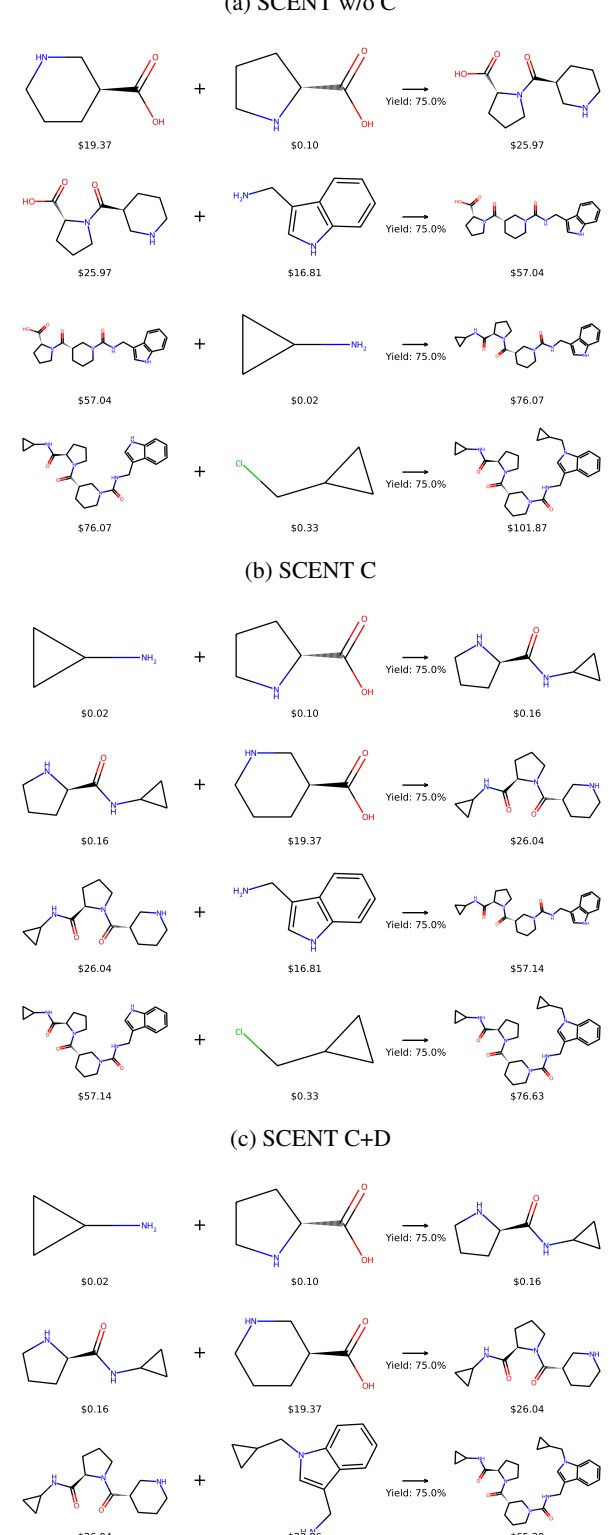

Figure 10: Three example pathways from SMALL setting leading to the same molecule. SCENT (C) is cheaper than SCENT (w/o C) by introducing the costly fragment ($16.81) later in the pathway. SCENT (C+D) further lowers cost via a convergent strategy that shortens the longest linear sequence (LLS), boosting overall yield.

Table 5: Evaluation of different template-based approaches using JNK3 proxy in SMALL, MEDIUM and LARGE settings. In all metrics, except "reward", we consider molecules with reward $> 0.8$. We observe that SCENT (C) that uses Cost Guidance can greatly reduce the cost of the sampled molecules in SMALL and MEDIUM settings, while drastically increasing the number of scaffolds.

| | model | online discovery | | inference sampling | |
|---|---|---|---|---|---|
| | | modes $> 0.8$ ↑ | scaff. $> 0.8$ ↑ | reward ↑ | cost $> 0.8$ ↓ |
| **SMALL** | RGFN | **307** ±**11** | **2852** ±**42** | **0.8** ±**0.0** | 11.6 ±0.3 |
| | SynFlowNet | 0.0 ±0.0 | 0.0 ±0.0 | 0.42 ±0.08 | - |
| | RxnFlow | 16.3 ±1.9 | 70.3 ±4.1 | 0.69 ±0.0 | 12.2 ±0.2 |
| | SCENT (w/o C) | 271 ±9 | 2363 ±100 | 0.78 ±0.01 | 16.8 ±2.8 |
| | SCENT (C) | 256 ±11 | 2248 ±138 | **0.8** ±**0.01** | 9.27 ±0.5 |
| | SCENT (D) | 287 ±9 | 2708 ±38 | **0.78** ±**0.0** | 11.7 ±0.6 |
| | SCENT (P) | 264 ±6 | 2441 ±95 | **0.79** ±**0.01** | 14.2 ±0.7 |
| | SCENT (C+D) | 293 ±1 | 2771 ±38 | 0.79 ±0.0 | **8.82** ±**0.36** |
| | SCENT (C+P) | 247 ±12 | 2395 ±191 | **0.8** ±**0.02** | 12.0 ±3.4 |
| | SCENT (C+D+P) | 303 ±6 | 2765 ±150 | **0.8** ±**0.0** | 9.45 ±0.55 |
| **MEDIUM** | RGFN | **737** ±**62** | 5685 ±969 | 0.6 ±0.01 | 1699 ±83 |
| | SynFlowNet | 0.33 ±0.47 | 0.33 ±0.47 | 0.5 ±0.0 | - |
| | RxnFlow | 4.0 ±1.0 | 4.0 ±1.0 | 0.51 ±0.01 | - |
| | SCENT (w/o C) | 711 ±75 | 11383 ±5011 | 0.62 ±0.03 | 1599 ±248 |
| | SCENT (C) | 525 ±61 | 51264 ±23502 | 0.77 ±0.02 | 1197 ±320 |
| | SCENT (D) | 875 ±138 | 13497 ±1009 | 0.65 ±0.0 | 1468 ±333 |
| | SCENT (P) | 836 ±133 | 13778 ±3952 | 0.65 ±0.01 | 1539 ±149 |
| | SCENT (C+D) | 423 ±108 | 60061 ±31036 | **0.81** ±**0.01** | 1063 ±352 |
| | SCENT (C+P) | 637 ±32 | **91872** ±**5942** | 0.76 ±0.01 | **869** ±**38** |
| | SCENT (C+D+P) | 645 ±330 | 89508 ±53412 | 0.79 ±0.06 | 1909 ±395 |
| **LARGE** | RGFN | 203 ±7 | 305 ±14 | 0.63 ±0.02 | 1652 ±59 |
| | SynFlowNet | 5.0 ±7.07 | 83.3 ±117.9 | 0.51 ±0.1 | - |
| | RxnFlow | 3.0 ±1.63 | 3.67 ±2.49 | 0.5 ±0.01 | - |
| | SCENT (w/o C) | 430 ±295 | 1817 ±1219 | 0.58 ±0.01 | 2454 ±637 |
| | SCENT (C) | 116 ±44 | 3165 ±2437 | 0.67 ±0.02 | **1213** ±**186** |
| | SCENT (D) | 112 ±129 | 502 ±348 | 0.57 ±0.02 | - |
| | SCENT (P) | 99 ±93.5 | 536 ±298 | 0.57 ±0.03 | 2531 ±563 |
| | SCENT (C+D) | 113 ±87 | 10412 ±5619 | **0.71** ±**0.01** | 1403 ±405 |
| | SCENT (C+P) | 442 ±325 | **15192** ±**9772** | **0.71** ±**0.04** | 1383 ±126 |
| | SCENT (C+D+P) | **472** ±**238** | 11616 ±5744 | **0.71** ±**0.01** | 1278 ±43 |

Table 6: GPU runtimes (in minutes) for all template-based baselines on the sEH proxy, averaged over 3 random seeds using a V100 32GB GPU.

| Model | SMALL | MEDIUM | LARGE |
|---|---|---|---|
| RGFN | 2882 ± 115 | 3892 ± 153 | 4455 ± 392 |
| SynFlowNet | 200 ± 3 | 2350 ± 162 | 8416 ± 19 |
| RxnFlow | 891 ± 4 | 3304 ± 44 | 3399 ± 50 |
| SCENT (C+D+P) | 800 ± 15 | 2849 ± 144 | 3688 ± 317 |

the trajectories and the average price of the fragments used also in the SMALL setting. Interestingly, it does not significantly limit the usage of fragments, which is likely due to their shortage and relatively uniform cost. We also see that the reaction yield does not play a role in reducing the cost.

# I  Further Comparison to Baselines

In this section, we extend our comparison of SCENT to established baseline models from the literature: REINVENT [20] and Graph GA [53]. Figure 13 shows a comparative analysis of the

(a) SCENT w/o C

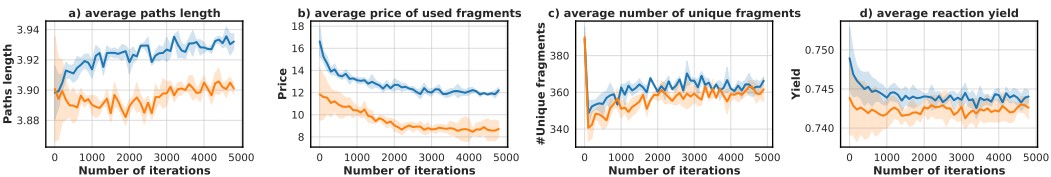

(b) SCENT C

Figure 11: Two example pathways leading to structurally similar high-rewarded molecules. A molecule proposed by SCENT (C) uses much cheaper fragment ($27.27 vs $279.35).

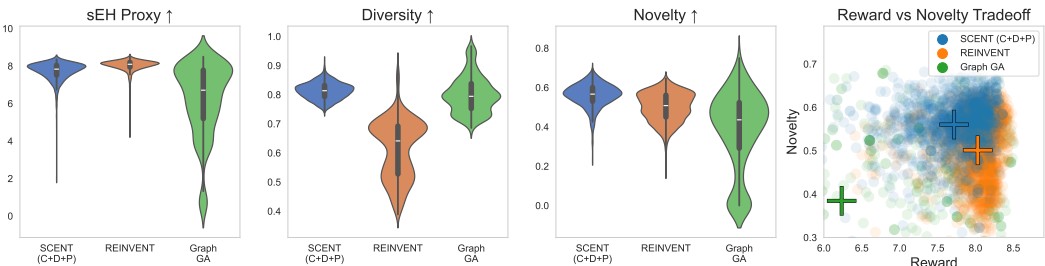

Figure 12: Cost reduction analysis for sEH proxy and SMALL setting. Cost Guidance decreases the average length of the sampled trajectory (a), and the average cost of used fragments (b) which directly reduces the synthesis path cost. Importantly, the average reaction yield (d) does not increase when using Cost Guidance, suggesting its low influence in reducing the synthesis costs. The plots are smoothened by averaging results from the last 100 iterations.

molecular properties generated by each method, filtered through AIZynthFinder [48] to retain only retrosynthetically accessible compounds. SCENT achieves a reward comparable to REINVENT, while outperforming it in terms of diversity and novelty. Graph GA generates highly diverse molecules, but these are often challenging to synthesize, as indicated by its low support in the final sub-plot. Importantly, both SCENT and REINVENT lie on the Pareto front of the reward–novelty trade-off: SCENT is biased toward higher novelty, whereas REINVENT tends toward higher reward. Table 7 and 8 further support this comparison, confirming that SCENT generates diverse, high-quality, and synthetically accessible molecules.

Figure 13: Comparison of molecules sampled by SCENT, REINVENT, and Graph GA, filtered using AIZynthFinder to retain only synthetically accessible candidates. Diversity is measured as the average Tanimoto distance between a molecule and the rest of the sampled set, while novelty is defined as the average Tanimoto distance to the closest molecule in the ChEMBL database. SCENT was trained in the SMALL setting. We observe that it generates diverse, high-reward, and synthesizable molecules, and lies on the Pareto front of the novelty–reward trade-off. Tick marks in the final sub-plots indicate the mean values of the sampled distributions.

Table 7: Comparison of generative models. Diversity, Novelty, and AIZynth scores were computed on the top 100 highest-scoring molecules. Diversity is defined as the average Tanimoto distance between each molecule and the rest of the sampled set. Novelty refers to the average Tanimoto distance to the closest molecule in the ChEMBL database. AIZynth represents the percentage of molecules successfully retrosynthesized by the AIZynthFinder software. All results are averaged over three independent runs of training and evaluation. Template-based models were trained under the SMALL setting.

| | Setting | Reward ↑ | Diversity ↑ | Novelty ↑ | AIZynth ↑ |
|---|---|---|---|---|---|
| Template-based | RGFN | $7.34_{\pm 0.06}$ | $0.77_{\pm 0.02}$ | $0.58_{\pm 0.01}$ | $0.647_{\pm 0.037}$ |
| | SynFlowNet | $6.89_{\pm 0.29}$ | $0.74_{\pm 0.02}$ | $0.53_{\pm 0.01}$ | $0.247_{\pm 0.090}$ |
| | RxnFlow | $6.30_{\pm 0.03}$ | $\mathbf{0.80}_{\pm \mathbf{0.00}}$ | $0.53_{\pm 0.01}$ | $0.653_{\pm 0.021}$ |
| | SCENT (C+D+P) | $\mathbf{7.66}_{\pm \mathbf{0.03}}$ | $0.73_{\pm 0.01}$ | $\mathbf{0.59}_{\pm \mathbf{0.01}}$ | $\mathbf{0.773}_{\pm \mathbf{0.066}}$ |
| Unconstrained | Graph GA | $7.30_{\pm 0.31}$ | $0.56_{\pm 0.05}$ | $0.62_{\pm 0.08}$ | $0.113_{\pm 0.160}$ |
| | FGFN | $6.03_{\pm 0.10}$ | $\mathbf{0.85}_{\pm \mathbf{0.00}}$ | $\mathbf{0.66}_{\pm \mathbf{0.00}}$ | $0.010_{\pm 0.000}$ |
| | FGFN + SA | $6.16_{\pm 0.20}$ | $0.80_{\pm 0.01}$ | $0.54_{\pm 0.02}$ | $0.453_{\pm 0.091}$ |
| | REINVENT | $\mathbf{7.97}_{\pm \mathbf{0.07}}$ | $0.59_{\pm 0.10}$ | $0.50_{\pm 0.05}$ | $\mathbf{0.800}_{\pm \mathbf{0.113}}$ |

Table 8: Comparison of generative models on Quantitative Estimation of Drug-likeness (QED), weight, Synthetic Complexity (SC), and Synthetic Accessibility (SA) metrics. Template-based models were trained on SMALL setting.

| | Setting | QED ↑ | weight ↓ | SC ↓ | SA ↓ |
|---|---|---|---|---|---|
| Template-based | RGFN | $0.244_{\pm 0.003}$ | $542_{\pm 7}$ | $4.84_{\pm 0.01}$ | $3.30_{\pm 0.03}$ |
| | SynFlowNet | $\mathbf{0.454}_{\pm \mathbf{0.113}}$ | $\mathbf{451}_{\pm \mathbf{76}}$ | $4.42_{\pm 0.28}$ | $\mathbf{3.06}_{\pm \mathbf{0.11}}$ |
| | RxnFlow | $0.259_{\pm 0.007}$ | $603_{\pm 9}$ | $\mathbf{3.34}_{\pm \mathbf{0.04}}$ | $4.73_{\pm 0.06}$ |
| | SCENT (C+D+P) | $0.286_{\pm 0.004}$ | $519_{\pm 5}$ | $4.82_{\pm 0.01}$ | $3.24_{\pm 0.02}$ |
| Unconstrained | Graph GA | $\mathbf{0.464}_{\pm \mathbf{0.033}}$ | $\mathbf{367}_{\pm \mathbf{12}}$ | $\mathbf{3.72}_{\pm \mathbf{0.23}}$ | $3.55_{\pm 0.66}$ |
| | FGFN | $0.197_{\pm 0.003}$ | $668_{\pm 5}$ | $4.88_{\pm 0.01}$ | $4.77_{\pm 0.06}$ |
| | FGFN + SA | $0.186_{\pm 0.004}$ | $621_{\pm 10}$ | $4.77_{\pm 0.04}$ | $3.42_{\pm 0.12}$ |
| | REINVENT | $0.188_{\pm 0.088}$ | $655_{\pm 179}$ | $4.64_{\pm 0.07}$ | $\mathbf{2.76}_{\pm \mathbf{0.12}}$ |

## J   Ablations

### J.1   Decomposability Guidance Ablation

Decomposability Guidance, described in Section 3.2.2, is an integral part of our SCENT model. Leveraging a machine learning model, it predicts which molecules can be further decomposed in the backward transition (retrosynthesized) and discards the transitions that would otherwise lead to invalid states. Table 9 shows that this component is crucial for the performance of SCENT.

Table 9: Effect of Decomposability guidance on online discovery and inference in SCENT.

| | model | online discovery | | inference sampling | |
|---|---|---|---|---|---|
| | | modes > 8.0 ↑ | scaff. > 8.0 ↑ | reward ↑ | cost > 8.0 ↓ |
| SMALL | w/o Decomposability | $309_{\pm 4}$ | $2794_{\pm 77}$ | $7.3_{\pm 0.03}$ | $52.2_{\pm 4.9}$ |
| | w/ Decomposability | $\mathbf{510}_{\pm \mathbf{26}}$ | $\mathbf{5413}_{\pm \mathbf{334}}$ | $\mathbf{7.38}_{\pm \mathbf{0.02}}$ | $\mathbf{37.1}_{\pm \mathbf{1.0}}$ |
| MEDIUM | w/o Decomposability | $6989_{\pm 303}$ | $8840_{\pm 652}$ | $7.15_{\pm 0.05}$ | $1549_{\pm 81}$ |
| | w/ Decomposability | $\mathbf{9310}_{\pm \mathbf{863}}$ | $\mathbf{11478}_{\pm \mathbf{823}}$ | $\mathbf{7.31}_{\pm \mathbf{0.09}}$ | $\mathbf{1463}_{\pm \mathbf{62}}$ |

Table 10: The machine learning-based cost estimation significantly outperforms constant value cost estimation, providing a substantial performance boost, particularly in cost reduction.

| | model | online discovery | | inference sampling | |
| --- | --- | --- | --- | --- | --- |
| | | modes > 8.0 ↑ | scaff. > 8.0 ↑ | reward ↑ | cost > 8.0 ↓ |
| SMALL | w/o Cost Guidance | 510 ±26 | 5413 ±334 | 7.38 ±0.02 | 37.1 ±1.0 |
| | Constant | **515** ±**9** | **5259** ±**150** | **7.43** ±**0.07** | 45.7 ±2.8 |
| | ML predicted | 478 ±11 | 5150 ±87 | **7.43** ±**0.02** | **23.7** ±**4.0** |
| MEDIUM | w/o Cost Guidance | 9310 ±863 | 11478 ±823 | 7.31 ±0.09 | 1463 ±62 |
| | Constant | 13121 ±1073 | 25487 ±907 | 7.41 ±0.02 | 1368 ±149 |
| | ML predicted | **17705** ±**4224** | **52340** ±**4303** | **7.74** ±**0.04** | **1163** ±**147** |
| LARGE | w/o Cost Guidance | 7171 ±291 | 8767 ±429 | 7.13 ±0.12 | 1678 ±63 |
| | Constant | **12218** ±**1725** | 26348 ±5397 | 7.44 ±0.12 | 1369 ±127 |
| | ML predicted | 11730 ±1777 | **88592** ±**16256** | **7.84** ±**0.06** | **1014** ±**137** |

## J.2 Synthesibility Cost Guidance Ablations

### J.2.1 Cost Prediction Model Ablation

In this section, we demonstrate the importance of accurately estimating the true cost $c$ using the machine learning model $c'$ for the optimal performance of SCENT. We compare our model with a low-quality cost estimator that outputs a constant value for every input molecule, where this constant is the mean of all the building blocks in the given scenario. It is important to note that we only modify the recursive component of the $C^S$ function from Equation (5), keeping the fragment costs unchanged. As shown in Table 10, the use of a constant value can reduce costs and improve SCENT's performance in MEDIUM and LARGE settings. However, machine learning-based cost estimation significantly outperforms this approach, providing a substantial performance boost, particularly in cost reduction. From this, we conclude that the quality of the cost prediction model used in Cost Guidance is critical to the performance of SCENT.

### J.2.2 Comparison of Cost-Reducing Methods

In this section, we compare our cost-guided SCENT with other methods to reduce the synthesis cost. The first method is to use the forward synthesis cost as a reward function component (we denote this approach as the "cost reward"). The second method that we tested is trained on subsets of the building block library $M$ with the top $n\%$ of the most expensive molecules being removed. While both approaches can effectively reduce the cost of proposed molecules, they lag behind cost-guided SCENT in terms of online discovery metrics, as can be seen in Figure 14. This suggests that cost biasing strikes the perfect balance between cost-effectiveness and diversity as it is able to consistently find more modes.

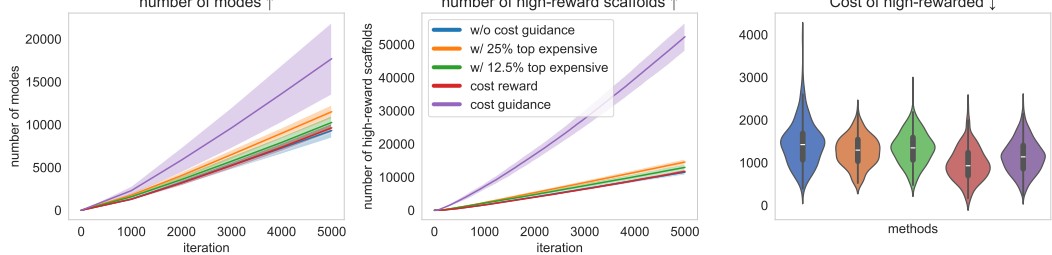

Figure 14: Plot comparing different approaches to reduce the cost of the molecules in MEDIUM setting on sEH proxy. We observe that Cost Guidance discovers by far the largest number of (high-reward) modes and scaffolds while reducing the cost.

### J.2.3 Update of Building Block Costs

We conducted an experiment on SCENT (C+D+P) to evaluate system robustness to changes in fragment costs.

At iteration 3000, we randomly reshuffled fragment costs and continued training without resetting model parameters, only updating the costs estimates used to train the cost predictor. Plots showing the evolution of cost and reward over time can be found in Figure 15. As expected, the trajectory cost spiked immediately after the change but quickly decreased, converging to the same level reached by a model trained from scratch with the new cost assignments (within 1000 iterations). The conclusion is that SCEN (C+D+P) does not require full retraining when costs change. Updating the cost predictor is sufficient.

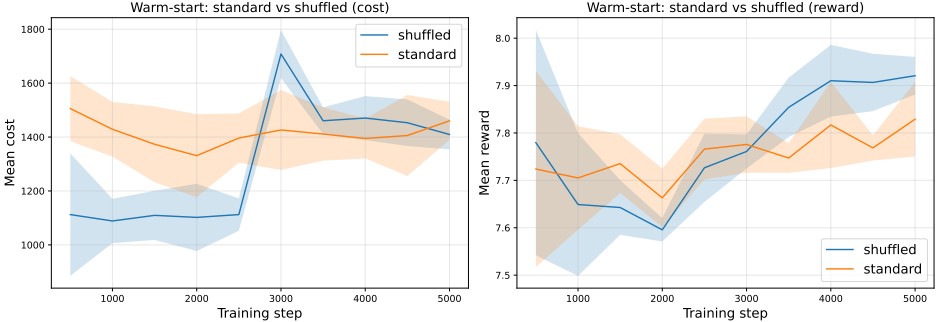

Figure 15: The evolution of cost and reward over training iterations. "Standard" refers to the method where costs were shuffled at the start of training. "Shuffled" refers to the method where shuffling occurred at the 3000th iteration. Results are reported on 1500 validation trajectories and averaged across 3 random seeds.

### J.3 Dynamic Library Ablation

Figure 16 illustrates how the performance of the reward-based Dynamic Library scales with the size of the initial set of building blocks. The improvement over the baseline (SCENT (C+P)) is substantial when the initial library is small but diminishes as its size increases. We also conduct an ablation study on the library-building mechanism, comparing our method—selecting the top $L$ molecules from the reward-sorted buffer—to a variant where $L$ molecules are randomly sampled from the buffer. Our results indicate that the selection mechanism plays a crucial role and is the primary driver of performance improvements.

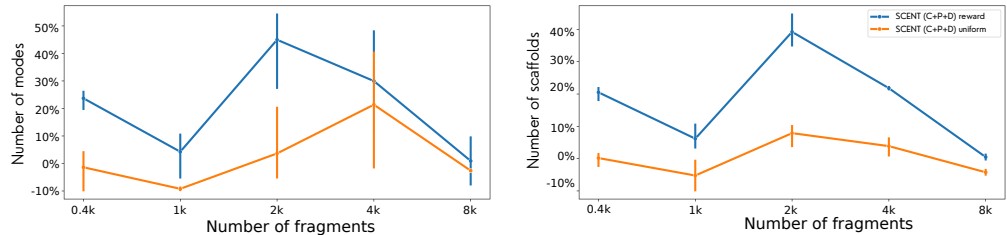

Figure 16: Plot comparing the relative improvement in performance between using a reward-based Dynamic Library (SCENT (C+P+D)) as opposed to a uniform-based one over the baseline (SCENT (C+P)).

### J.4 Exploitation Penalty vs Other Exploration Techniques

In this section, we compare our Exploitation Penalty to the widely used $\epsilon$-greedy exploration where $\epsilon = 0.05$. Additionally, we "decay" variants that linearly decrease the exploration parameters ($\epsilon$ for $\epsilon$-greedy, and $\gamma$ for exploitation penalty) so that they reach 0 at the 1000-th iteration. Figure 17

compares the exploration techniques on three variations of SCENT: with/without Cost Guidance, and with Dynamic Library. We observe that exploitation penalty with linear decaying (penalty decaying) helps our cost guided SCENT the most, successfully mitigating its exploitative nature. Dynamic Library also benefits from using an exploitation penalty, possibly because the penalty explicitly encourages the model to visit newly added molecules. Table 11 shows that our Exploitation Penalty helps the cost-guided SCENT in exploration, especially when accompanied with Dynamic Library.

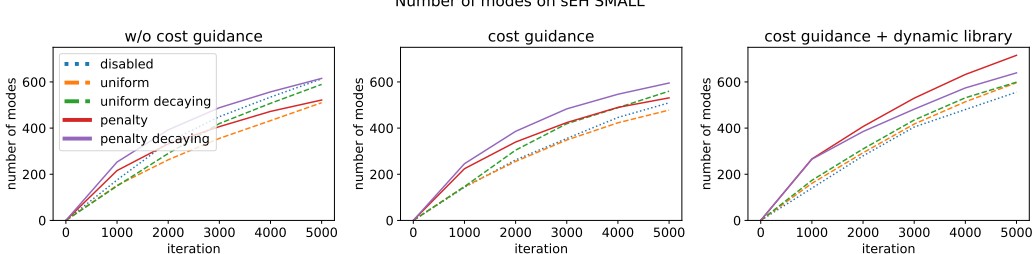

Figure 17: Comparison of the exploration techniques on three variations of SCENT: with/without Cost Guidance, and with Dynamic Library. We observe that exploitation penalty with linear decaying (penalty decaying) helps our cost guided SCENT while Dynamic Library benefits significantly from using a non-decaying penalty.

Table 11: Comparison of different exploration techniques for sEH in SMALL setting.

| | model | online discovery | | inference sampling | |
| --- | --- | --- | --- | --- | --- |
| | | modes > 8.0 ↑ | scaff. > 8.0 ↑ | reward ↑ | cost > 8.0 ↓ |
| w/o cost guidance | disabled | 613 ±20 | 7599 ±233 | 7.42 ±0.05 | 39.3 ±3.9 |
| | uniform | 510 ±26 | 5413 ±334 | 7.38 ±0.02 | 37.1 ±1.0 |
| | uniform decaying | 589 ±12 | 6880 ±386 | 7.41 ±0.06 | **33.2** ±**3.9** |
| | penalty | 521 ±15 | 3660 ±99 | **7.47** ±**0.02** | 39.6 ±2.1 |
| | penalty decaying | **615** ±**16** | **7603** ±**339** | 7.38 ±0.02 | 39.4 ±2.4 |
| cost guidance | disabled | 509 ±24 | 5794 ±337 | 7.4 ±0.08 | 23.9 ±7.6 |
| | uniform | 478 ±11 | 5150 ±87 | 7.43 ±0.02 | 23.7 ±4.0 |
| | uniform decaying | 560 ±3 | 6487 ±299 | 7.43 ±0.04 | 21.6 ±1.4 |
| | penalty | 531 ±31 | 4162 ±279 | **7.62** ±**0.01** | 26.3 ±4.0 |
| | penalty decaying | **595** ±**20** | **6839** ±**360** | 7.42 ±0.05 | **20.7** ±**2.6** |
| cost guidance + dyn. library | disabled | 555 ±43 | 7098 ±770 | 7.56 ±0.04 | 20.5 ±2.6 |
| | uniform | 596 ±19 | 7148 ±497 | 7.56 ±0.05 | **19.7** ±**2.0** |
| | uniform decaying | 599 ±19 | 7670 ±334 | 7.49 ±0.07 | 21.0 ±0.9 |
| | penalty | **715** ±**15** | **8768** ±**363** | **7.66** ±**0.03** | 20.7 ±3.0 |
| | penalty decaying | 639 ±8 | 8323 ±276 | **7.66** ±**0.01** | 21.0 ±1.2 |

