# OpenReview forum: "Scalable and Cost-Efficient de Novo Template-Based Molecular Generation"
_NeurIPS.cc/2025/Conference — NeurIPS 2025 poster_

### Official Review · Reviewer_C46i · 2025-07-02

**Clarity:** 2
**Significance:** 4
**Originality:** 3
**Rating:** 5
**Confidence:** 3

**Summary:**

The paper introduces SCENT, a novel framework for template-based molecular generation using GFlowNets. SCENT addresses key challenges in reaction-based GFlowNets by incorporating Recursive Cost Guidance, which leverages ML models to guide the generation process toward lower synthesis cost and valid backward transitions. The method further introduces an Exploitation Penalty to promote exploration and a Dynamic Library mechanism that reuses promising intermediate molecules, enabling full-tree synthesis and broader mode coverage.

**Questions:**

1. Could the authors provide a comparison of the performance of SCENT under the same training conditions with random cropping of the number of available fragments? This question follows from the statement “Previous work shows that reducing library size via random cropping can help, but the effect is modest compared to our gains” (L287-288).
2. In Figure 6b, the number of modes for SCENT (C) and SCENT (C+P) seems to narrow at M=128k compared to M=64k. Could the authors elaborate on why this narrowing occurs despite having more building blocks?
3. Could the authors provide a brief characterization of datasets (SMALL, MEDIUM, LARGE) beyond just size? For example, are there differences in molecular complexity, diversity, or chemical space coverage between the curated SMALL set and the larger Enamine sets? What supports the claim of "broad coverage of molecular space" (L587)?

**Ethical Concerns:**

["NO or VERY MINOR ethics concerns only"]

**Final Justification:**

The authors have addressed my concerns in their rebuttal, including clarifications on computational resources, cost model evaluation, dataset characterization, and limitations. The new details strengthen the paper’s clarity and reproducibility. While some minor issues remain, these do not substantially diminish from the contribution.

**Limitations:**

The authors point out some limitations of the model in the text of the article, but it would be more useful to form a separate section in which the following points can be discussed:
1. The paper focuses on synthesizability and cost, which are crucial, but does not include any assessment of the toxicity of the generated molecules. While outside the direct scope of the method, it's a critical aspect for drug discovery.
2. The performance of SCENT (and baselines) shows variability across different tasks (sEH, GSK3β, JNK3). This suggests that the inherent suitability of the method might depend on the specific target, which could be a limitation in real-world applications.
3. While the paper addresses synthesis cost, real-world synthesis complexity involves many factors beyond just building block cost and reaction yield L576-578 (e.g., purification challenges, hazardous reagents, reaction conditions, scalability). The current cost model is a first-order approximation (L151-155), and acknowledging these additional complexities would provide a more complete picture.

**Paper Formatting Concerns:**

There are no major formatting issues.

**Quality:**

3

**Strengths And Weaknesses:**

**Strengths:**
1. The proposed components (Recursive Cost Guidance, Exploitation Penalty, Dynamic Library) are clearly articulated and logically integrated into the GFlowNet framework, demonstrating a cohesive design.
2. The paper provides comprehensive ablation studies that effectively demonstrate the individual contributions and synergistic effects of each proposed component, supporting the design choices.
3. SCENT is rigorously compared against a strong set of relevant baselines (RGFN, SynFlowNet, RxnFlow, REINVENT, Graph GA) across multiple tasks and library sizes, providing convincing evidence of its superior performance.
4. By focusing on synthesizability and cost-efficiency, SCENT directly addresses practical challenges in drug discovery, making generated molecules more useful for experimental use.

**Weaknesses:**
1. The paper lacks specific estimates of computational resources (e.g., training time on a particular GPU, memory usage) required for training SCENT, especially in comparison to baselines like RGFN. This information is crucial for reproducibility and practical adoption.
2. While the sEH proxy task is mentioned, a brief explanation of what it measures, how its reward is calculated, and what a high score signifies would greatly enhance accessibility for readers less familiar with specific drug discovery benchmarks.
3. The paper describes the architecture and training of the Synthesis Cost Model and Decomposability Model but does not provide performance metrics (e.g., R2 score for regression, accuracy/F1 score for classification) for these auxiliary models.
4. The paper mentions Tanimoto similarity for the selection of molecules. However, it now seems more appropriate to use more modern metrics (e.g., MCES [1])  for assessing molecular similarity, which mitigate some of the limitations of Tanimoto similarity.

[1] Kretschmer, F., Seipp, J., Ludwig, M., Klau, G. W., & Böcker, S. (2025). Coverage bias in small molecule machine learning. Nature Communications, 16(1), 554.

**Minor comments:**
1. A single, comprehensive diagram illustrating the entire SCENT pipeline, showing how the forward policy, backward policy, cost guidance, exploitation penalty, and dynamic library interact, would significantly improve clarity and understanding. Figure 1 is adequate but focuses on individual components rather than the full system flow.
2. The paper sometimes uses "exploration" and "exploitation" in a way that could be slightly confusing (e.g., L309, L676 - “Exploration Penalty”, L10 and others - “Exploitation Penalty”).
3. The labels "w/" in Figure 10 are ambiguous as it is stated in text that “the top n% of the most expensive molecules being removed.”

---

> ### Author Rebuttal · Authors · 2025-07-31
>
> Thank you very much for your thoughtful and constructive review. We’ve carefully addressed all the issues you raised and have implemented your suggestions related to computational resource metrics and cost model evaluation. In particular, the evaluation of the Synthesis Cost Guidance is now complete, and we are actively working on the analysis of the Decomposability Model.
>
> We believe these enhancements have significantly strengthened the paper and trust that the revisions will provide additional clarity and depth to our contributions.
>
> ## Experiments with random cropping
>
> As shown in Figure 6, reducing the fragment library size in SCENT (w/o C) generally leads to decreased or stagnant performance. Since the smaller libraries are strict and random subsets of the larger ones, this reduction effectively “crops” the available fragments. By contrast, adding cost biasing (C) and exploitative penalty (P) yields significant performance gains.
>
> From these results, we conclude that expanding the fragment library is more beneficial than cropping it, particularly when combined with cost biasing and exploitative penalty. The largest improvements are observed for SCENT (C+P), which underscores the importance of maintaining a rich and diverse fragment library.
>
> ## Narrowed number of modes in 128k
>
> Thank you for the insightful question. As the number of building blocks increases, the state space grows exponentially, which makes learning more challenging under a fixed training iteration budget. This may explain why the number of modes narrows at M=128k, despite having more fragments available. This trend is consistent with prior observations in related work (e.g., RGFN). Notably, SCENT is able to maintain strong performance even at higher fragment counts, significantly extending the range over which improvements are observed compared to RGFN.
>
> ## Building block libraries analysis
>
> Thank you for your helpful question. To characterize the SMALL, MEDIUM, and LARGE building blocks libraries beyond their size, we analyzed molecular diversity, complexity, and key physicochemical properties relevant to drug discovery. Specifically, we computed:
>
> - **Diversity:** Number of unique molecular modes and scaffolds, using leader clustering (with a Tanimoto similarity threshold of 0.3) to define clusters.
> - **Complexity:** Metrics including molecular weight and rotatable bonds.
> - **Physicochemical features:** Properties such as logP, hydrogen bond donors (HBD), hydrogen bond acceptors (HBA), and topological polar surface area (TPSA), which influence solubility, permeability, and binding potential.
>
> The summary statistics are as follows:
>
> | Property | SMALL | MEDIUM | LARGE | Interpretation |
> | --- | --- | --- | --- | --- |
> | Number of modes | 154 | 9,738 | 16,181 | Molecular diversity |
> | Number of scaffolds | 39 | 8,118 | 16,149 | Scaffold diversity (core chemotypes) |
> | Molecular Weight (Da) | 150 ± 44 | 212 ± 62 | 221 ± 68 | Molecular complexity |
> | Rotatable Bonds | 1.23 ± 0.98 | 2.09 ± 1.61 | 2.34 ± 1.69 | Molecular flexibility/complexity |
> | logP | 0.96 ± 1.25 | 1.71 ± 1.18 | 1.80 ± 1.22 | Lipophilicity / solubility balance |
> | HBD | 1.12 ± 1.06 | 1.06 ± 0.76 | 1.07 ± 0.77 | Hydrogen bond donors (binding potential) |
> | HBA | 1.97 ± 0.84 | 2.77 ± 1.23 | 2.89 ± 1.31 | Hydrogen bond acceptors |
> | TPSA (Å²) | 41.0 ± 23.4 | 49.9 ± 22.9 | 51.0 ± 23.8 | Polar surface area (transport/permeability) |
>
> These data show that the curated SMALL dataset represents a focused set of chemically simpler molecules with fewer scaffolds and lower molecular weight, while the MEDIUM and LARGE datasets encompass increasingly diverse chemical scaffolds and more complex molecules. The LARGE set in particular covers a broad chemical space, as indicated by the large number of unique scaffolds and modes and the wider range of physicochemical properties.
>
> This comprehensive characterization supports our claim of “broad coverage of molecular space” by demonstrating substantial diversity in core scaffolds and molecular features critical to drug discovery across the datasets.
>
> ## Computational resources
>
> As requested by the reviewers, we report GPU runtimes (in minutes) for all baselines on the sEH proxy, averaged over 3 random seeds using a V100 32GB GPU. All methods used a batch size of 64, except SynFlowNet, which required batch size 32 on the LARGE setting even after optimizing its policy code to fit within memory limits.
>
> | Model | SMALL | MEDIUM | LARGE |
> | --- | --- | --- | --- |
> | RGFN | 2882 ± 115 | 3892 ± 153 | 4455 ± 392 |
> | SynFlowNet | 200 ± 3 | 2350 ± 162 | 8416 ± 19 |
> | RxnFlow | 891 ± 4 | 3304 ± 44 | 3399 ± 50 |
> | SCENT (C+D+P) | 800 ± 15 | 2849 ± 144 | 3688 ± 317 |
>
> RGFN shares a part of the codebase with SCENT, allowing for direct comparison. We observe consistent improvements across all settings, particularly in the SMALL configuration. While SynFlowNet is faster on SMALL and MEDIUM, it struggles on LARGE, even with our memory optimizations. However, differences in codebases make direct comparisons less definitive.
>
> Note that our implementation is not optimized for runtime and could benefit from further improvements.
>
> ## Explain sEH
>
> Following the suggestion of the reviewer we added a brief explanation of sEH proxy to the revised paper: “The **sEH proxy** is a graph neural network  trained to predict the binding affinity of small molecules to **soluble epoxide hydrolase (sEH)**, using docking scores computed by AutoDock Vina as training labels. The higher the reward defined by sEH proxy, the more strongly a given molecule binds to sEH.”
>
> ## Performance of cost models
>
> Thank you for the valid suggestion. In response to Reviewer QKfB, we conducted an analysis of the cost model used in Synthesis Cost Guidance (see *Performance of Cost Models*). A similar analysis for the Decomposability Cost Guidance is currently in progress, and we will provide the results as soon as they become available.
>
> ## MCES instead of Tanimoto
>
> Thank you for drawing our attention to MCES. We agree that it can offer a more nuanced measure of molecular similarity compared to Tanimoto similarity. However, we opted for Tanimoto primarily because (1) it is widely used in related work, (2) it is significantly faster to compute.
>
> ## Limitation section
>
> We apologize for the lack of limitation section, we are going to include the following limitation summary in the revised paper:
>
> “While our study demonstrates the effectiveness of SCENT in generating cost-efficient and synthesizable molecules, several limitations remain.
>
> First, the synthesis cost estimates are simplified and heuristic, omitting factors such as solvent usage, purification steps, hazardous reagents, reaction conditions, and scalability. Consequently, the current cost model should be regarded as a first-order approximation. More accurate estimation would require lab-specific calibration, including robust yield prediction models and vendor-specific building block pricing.
>
> SCENT’s performance varies across different biological targets, indicating that its effectiveness may depend on task-specific factors. Identifying and addressing these bottlenecks is an important direction for future work. Key areas of improvement include refining the exploitation behavior of the Dynamic Library, and enhancing the generalizability of the cost predictor model used in Synthesis Cost Guidance. Additionally, we do not assess performance as the number of reaction templates increases, an important scalability consideration, as this expansion significantly enlarges the action space and introduces additional learning challenges.
>
> Our framework emphasizes synthesizability and cost, but does not incorporate toxicity or broader ADMET properties, which are critical for downstream drug development. Finally, we acknowledge the absence of wet-lab validation, which remains the definitive benchmark for evaluating the practical utility of our approach. We leave this to future work.”

---

> > ### Comment · Reviewer_C46i · 2025-08-04
> >
> > Thank you for your detailed and constructive rebuttal. Your responses have addressed my concerns, so I am raising my score to 5.

---

> > > ### Author Response · Authors · 2025-08-05
> > >
> > > We thank the reviewer for engaging with our rebuttal and positively assessing it. We have included a follow-up to the performance of cost models where we study the performance of decomposability models in the latest response to Reviewer QKfB.

---

### Official Review · Reviewer_E4oU · 2025-07-03

**Clarity:** 3
**Significance:** 3
**Originality:** 3
**Rating:** 4
**Confidence:** 4

**Summary:**

This paper presents SCENT, a template-based molecular generation framework using Generative Flow Networks. The authors propose three key components: Recursive Cost Guidance to estimate and minimize synthesis cost, an Exploitation Penalty to encourage exploration, and a Dynamic Library mechanism to improve synthesis diversity and scalability. SCENT is evaluated on multiple tasks and fragment library sizes, showing improvements over several baselines in terms of synthesis cost, diversity, and reward.

**Questions:**

1. The authors mention using stock prices and reaction yields to model synthesis cost. Could the authors clarify how these are obtained in non-SMALL settings (e.g., Enamine pricing and literature yield extrapolation)? And how robust are these estimates to inaccuracies or variability across different datasets or vendors?

2. Has SCENT been tested on a set of molecules with known real-world synthesis routes? Incorporating validation via retrosynthesis software (e.g., AiZynthFinder or ASKCOS) would strengthen the empirical claims.

3. While ablations are performed (Section 4.2), could the authors provide a more fine-grained breakdown of individual contributions of Recursive Cost Guidance vs. Dynamic Library vs. Exploitation Penalty in all tasks (sEH, JNK3, GSK3β)?

4. The performance of SCENT may depend on the hyperparameters of cost models (e.g., α, γ in penalty terms). Have the authors explored sensitivity analyses or automated tuning (e.g., Bayesian optimization)?

5. The backward policy guidance is central to SCENT. Could the authors visualize or analyze specific trajectories to illustrate how cost guidance steers generation differently from baselines?

**Ethical Concerns:**

["NO or VERY MINOR ethics concerns only"]

**Final Justification:**

Given the overall quality of the work and the improvements made in response to the review, I will maintain my positive rating.

**Limitations:**

The authors discuss some limitations, such as the approximations in cost modeling and the assumption of reaction availability. However, more explicit acknowledgment of the risks of using heuristic or curated data, and the limited external validation, would strengthen this section. Overall, the paper shows strong potential, but a few methodological clarifications and validations would make it more convincing.

**Paper Formatting Concerns:**

No.

**Quality:**

3

**Strengths And Weaknesses:**

**Strengths**

1. The Recursive Cost Guidance framework elegantly integrates cost estimation into the backward policy, offering a principled way to guide synthesis toward viable and low-cost molecules.

2. The authors conduct extensive comparisons across multiple baselines (RGFN, SynFlowNet, RxnFlow) and settings (SMALL, MEDIUM, LARGE).

3. SCENT consistently achieves better synthesis cost, reward, and diversity metrics, showing both practical and scientific utility.

4. The use of a Dynamic Library shows notable gains in large-scale fragment settings, addressing a major bottleneck in template-based approaches.

**Weaknesses**

However, some aspects of the method remain unclear. The estimation of synthesis cost and reaction yield is somewhat opaque and heavily relies on heuristics or external databases, which may limit reproducibility and generalizability. Additionally, the effectiveness of the Dynamic Library and cost models is not fully validated outside of synthetic benchmarks.

1. Some aspects of the cost prediction model (e.g., how exact yields and costs are estimated or normalized) are deferred to appendices and lack clarity in the main text.

2. The reaction yield estimation and building block cost model may not generalize well across domains, as the current implementation is based on vendor-specific and literature-curated data.

---

> ### Author Rebuttal · Authors · 2025-07-31
>
> Thank you for your thoughtful feedback. We have carefully addressed all the concerns raised and have incorporated your suggestion regarding a more fine-grained study of component contributions. The corresponding ablation experiments are underway, and we will include their results in the revised camera-ready version of the paper. We hope these additions may help to further clarify our approach and underscore our commitment to a thorough evaluation. We would greatly appreciate your kind consideration of these improvements when reassessing the paper.
>
> ## Yields and stock prices for MEDIUM and LARGE
>
> For the **MEDIUM** and **LARGE** settings, stock prices were obtained by automatically scraping Enamine’s publicly available catalog. These prices represent a static snapshot at the time of data collection and do not reflect temporal fluctuations or pricing differences across vendors. Yield estimates are derived from literature-based averages, which provide reasonable robustness across diverse reaction classes. However, as discussed in the *Cost Estimates Generalization* section, these estimates may lack precision in the context of specific experimental setups or lab conditions.
>
> ## Validation via retrosynthesis software
>
> SCENT is a de novo generative model that rarely reproduces molecules previously tested in the lab—typically, only ~1 in 150,000 generated compounds appears in ChEMBL. Thus, validating SCENT using retrosynthesis tools better reflects its intended use case than somehow testing it on a set of molecules with known synthesis routes. As shown in Figure 7, SCENT achieves a higher AiZynthFinder success rate than RGFN (likely due to its preference for shorter synthetic trajectories) and outperforms the non-template-based FGFN+SA, even though FGFN+SA is explicitly optimized for synthetic accessibility.
>
> ## Fine-grained components’ contributions
>
> We appreciate the suggestion and agree that a more fine-grained ablation would provide valuable insights. In response, we initiated experiments with SCENT (w/o C + P) and SCENT (w/o C + D) to better isolate the individual contributions of the Dynamic Library and the Exploitation Penalty across all tasks and settings.
>
> Unfortunately, these experiments are computationally intensive and have been delayed due to limited server availability, particularly during the rebuttal period, when compute resources are in high demand. While we are aiming to complete these runs before the end of the discussion phase on August 6th, we acknowledge that there is a chance they may not finish in time.
>
> Regardless, we are committed to including the full results in the camera-ready version of the paper and will notify the reviewers as soon as the results are available.
>
> ## Cost Guidance hyperparameters
>
> In the original paper, we have not explored the sensitivity, neither the automated tuning (we performed semi-manual small grid searches). Following the suggestion, we performed a detailed sensitivity analysis for each component (including temperature α for costs models, and γ in Exploitation Penalty). The results can be found in the response to Reviewer QKfB (Section *Hyperparameters*). Our results indicate that SCENT is generally resilient to variations in hyperparameters - its overall performance remains stable under a wide range of settings. That said, some hyperparameters have a notable impact on specific aspects of behavior; for example, changing $\alpha_D$ temperature for Decomposability Guidance can lead to a ~20% increase in the number of discovered modes. This highlights the potential for further improvements through systematic hyperparameter optimization using Bayesian optimization, which we leave as future work.
>
> ## Trajectory analysis
>
> As suggested by the reviewer, we conducted a detailed analysis of the generated synthesis trajectories and found that the Synthesis Cost Guidance module influences generation in two key ways: (1) it defers the use of expensive fragments to later stages in the trajectory, effectively reducing the quantity of such fragments needed to produce the same amount of final product; and (2) it favors cheaper fragment alternatives when available. This behavior aligns with the module’s objective of minimizing overall synthesis cost. A thorough analysis, along with concrete examples, is included in our response to Reviewer **9Lb3** (see *Pathways Analysis* section). We are unable to include figures or PDFs during the rebuttal, but we will incorporate the corresponding visualizations in the revised version of the paper.
>
> ## Cost estimates generalization
>
> We agree that the yield estimation and cost model may not generalize well across different labs. It is, however, a consequence of the inherent chemical synthesis variability and the lab-specific factors (e.g. contracted vendor) being inherently hard to account for. To our knowledge, there is no universally applicable yield estimation method [1], neither a gold-standard scheme of pricing the building blocks.
>
> Our framework is as general as it can be, in the sense that it can accommodate any cost estimation method, including those that are accurate yet very lab-specific.
>
> Moreover, in a study requested by Reviewer 9Lb3 (see *Warm-start study* section), we demonstrate that SCENT’s cost prediction module can effectively adapt to random perturbations in pricing during training. When price changes occur, whether due to market fluctuations or vendor contract updates, they can be directly incorporated into the model. SCENT is able to adapt to these changes within approximately 1,000 training iterations, eliminating the need for retraining from scratch each time prices are updated. This warm-start capability significantly improves the model’s practicality in dynamic pricing environments.
>
> [1] “When Yield Prediction Does Not Yield Prediction: An Overview of the Current Challenges”
>
> ## Costs and yield deferred to appendices
>
> Thank you for the comment, we considered this and would like to argue that the main focus of the paper is the generative framework, and the cost prediction model, while important for the framework to work, is in the end secondary. As such, we’d argue that keeping the details of this model in the appendix is appropriate. However, if the Reviewer feels strongly that shouldn’t be the case, we can incorporate additional details in the main body of the paper.
>
> ## Limitation section
>
> We apologize for the lack of limitation section, we are going to include the following limitation summary in the revised paper:
>
> “While our study demonstrates the effectiveness of SCENT in generating cost-efficient and synthesizable molecules, several limitations remain.
>
> First, the synthesis cost estimates are simplified and heuristic, omitting factors such as solvent usage, purification steps, hazardous reagents, reaction conditions, and scalability. Consequently, the current cost model should be regarded as a first-order approximation. More accurate estimation would require lab-specific calibration, including robust yield prediction models and vendor-specific building block pricing.
>
> SCENT’s performance varies across different biological targets, indicating that its effectiveness may depend on task-specific factors. Identifying and addressing these bottlenecks is an important direction for future work. Key areas of improvement include refining the exploitation behavior of the Dynamic Library, and enhancing the generalizability of the cost predictor model used in Synthesis Cost Guidance. Additionally, we do not assess performance as the number of reaction templates increases, an important scalability consideration, as this expansion significantly enlarges the action space and introduces additional learning challenges.
>
> Our framework emphasizes synthesizability and cost, but does not incorporate toxicity or broader ADMET properties, which are critical for downstream drug development. Finally, we acknowledge the absence of wet-lab validation, which remains the definitive benchmark for evaluating the practical utility of our approach. We leave this to future work.”

---

> > ### Author Response · Authors · 2025-08-04
> > **Fine-grained components’ contributions**
> >
> > We report the promised fine-grained ablation, highlighting the individual contributions of Exploitation Penalty (P) and Dynamic Library (D). The key conclusions, supported by the tables below, are as follows.
> >
> > Exploitation Penalty (P) and Dynamic Library (D) were tuned in the presence of Synthesis Cost Guidance (C), which likely results in their suboptimal performance as standalone modules. Despite this, both P and D demonstrate consistent improvements across all tasks in the SMALL setting when used alone. This aligns with intuition: in a constrained search space, models are more prone to over-exploitation of narrow patterns. Here, P explicitly discourages such behavior, while D expands the search space, facilitating broader exploration and solution diversity.
> >
> > In contrast, the individual effect of P and D is limited in MEDIUM and LARGE settings. This is likely because, in these larger action spaces, some degree of exploitation is beneficial and necessary to converge on strong solutions. Since P is inherently designed to reduce exploitation, and D extends the search space, their isolated application can counteract this need.
> >
> > However, when coupled with C, P and D become significantly more effective across all scales (as shown in the main results of the paper). C encourages exploitation through cost-based guidance, creating the conditions under which P and D can fully express their benefits. In essence, while these modules may not yield substantial gains in isolation in larger settings, they are complementary to C, reducing its over-exploitative tendencies. This synergy is particularly crucial, as the original motivation for introducing the Exploitation Penalty was precisely to mitigate the over-exploitation introduced by C.
> >
> > Below are the tables supporting the analysis. Results for SCENT (w/o C) and SCENT (C) were copied from the paper.
> >
> > ###sEH
> > | setting | model | #modes  | #scaffolds | reward  | cost |
> > | --- | --- | --- | --- | --- | --- |
> > | SMALL | SCENT (w/o C) | 510 ±26 | 5413 ±334 | 7.38 ±0.02 | 37.1 ±1.0 |
> > | SMALL | SCENT (C)  | 478 ±11 | 5150 ±87 | 7.43 ±0.02 | 23.7 ±4.0 |
> > | SMALL | SCENT (D) | 557 ± 6 | 6496 ± 34 | 7.42 ± 0.02 | 32.8 ± 1.9 |
> > | SMALL | SCENT (P) | 644 ± 6 | 7840 ± 169 | 7.33 ± 0.03 | 36.0 ± 1.4 |
> > | MEDIUM | SCENT (w/o C) | 9310 ±863 | 11478 ±823 | 7.31 ±0.09 | 1463 ±62 |
> > | MEDIUM | SCENT (C)  | 17705 ±4224 | 52340 ±4303 | 7.74 ±0.04 | 1163 ±147 |
> > | MEDIUM | SCENT (D) | 8629 ± 889 | 10605 ± 1034 | 7.21 ± 0.16 | 1522 ± 139 |
> > | MEDIUM | SCENT (P) | 13270 ± 647 | 17093 ± 570 | 7.32 ± 0.04 | 1462 ± 16 |
> > | LARGE | SCENT (w/o C) | 7171 ±291 | 8767 ±429 | 7.13 ±0.12 | 1678 ±63 |
> > | LARGE | SCENT (C)  | 12375 ±264 | 36930 ±5455 | 7.52 ±0.09 | 1267 ±159 |
> > | LARGE | SCENT (D) | 6384 ± 478 | 7956 ± 690 | 7.2 ± 0.11 | 1656 ± 40 |
> > | LARGE | SCENT (P) | 4594 ± 414 | 8475 ± 737 | 7.22 ± 0.05 | 1779 ± 85 |
> >
> > ###GSK
> > | setting | model | #modes  | #scaffolds | reward  | cost |
> > | --- | --- | --- | --- | --- | --- |
> > | SMALL | SCENT (w/o C) | 290 ±17 | 1911 ±84 | 0.7 ±0.01 | 44.1 ±5.1 |
> > | SMALL | SCENT (C)  | 287 ±9 | 1856 ±147 | 0.7 ±0.0 | 34.9 ±4.5 |
> > | SMALL | SCENT (D) | 319 ± 8 | 2196 ± 30 | 0.71 ± 0.01 | 38.0 ± 1.9 |
> > | SMALL | SCENT (P) | 330 ± 6 | 2447 ± 94 | 0.7 ± 0.01 | 45.8 ± 1.8 |
> > | MEDIUM | SCENT (w/o C) | 6430 ±2427 | 10010 ±2537 | 0.66 ±0.03 | 1177 ±187 |
> > | MEDIUM | SCENT (C)  | 5945 ±4009 | 50435 ±13360 | 0.74 ±0.03 | 917 ±200 |
> > | MEDIUM | SCENT (D) | 5668 ± 2149 | 8548 ± 2297 | 0.64 ± 0.02 | 1217 ± 90 |
> > | MEDIUM | SCENT (P) | 6494 ± 1812 | 11493 ± 1929 | 0.66 ± 0.02 | 1164 ± 39 |
> > | LARGE | SCENT (w/o C) | 3366 ±3503 | 4873 ±4494 | 0.6 ±0.07 | 1842 ±606 |
> > | LARGE | SCENT (C)  | 4295 ±1468 | 28867 ±2418 | 0.7 ±0.01 | 973 ±119 |
> > | LARGE | SCENT (D) | 2256 ± 472 | 3185 ± 514 | 0.57 ± 0.03 | 1390 ± 239 |
> > | LARGE | SCENT (P) | 1855 ± 234 | 2604 ± 314 | 0.6 ± 0.0 | 1376 ± 68 |
> >
> > ###JNK3
> >
> > | setting | model | #modes  | #scaffolds | reward  | cost |
> > | --- | --- | --- | --- | --- | --- |
> > | SMALL | SCENT (w/o C) | 271 ±9 | 2363 ±100 | 0.78 ±0.01 | 16.8 ±2.8 |
> > | SMALL | SCENT (C)  | 256 ±11 | 2248 ±138 | 0.8 ±0.01 | 9.27 ±0.5 |
> > | SMALL | SCENT (D) | 287 ± 9 | 2708 ± 38 | 0.78 ± 0.0 | 11.7 ± 0.6 |
> > | SMALL | SCENT (P) | 264 ± 6 | 2441 ± 95 | 0.79 ± 0.01 | 14.2 ± 0.7 |
> > | MEDIUM | SCENT (w/o C) | 711 ±75 | 11383 ±5011 | 0.62 ±0.03 | 1599 ±248 |
> > | MEDIUM | SCENT (C)  | 525 ±61 | 51264 ±23502  | 0.77 ±0.02 | 1197 ±320 |
> > | MEDIUM | SCENT (D) | 875 ± 138 | 13497 ± 1009 | 0.65 ± 0.0 | 1468 ± 333 |
> > | MEDIUM | SCENT (P) | 836 ± 133 | 13778 ± 3952 | 0.65 ± 0.01 | 1539 ± 149 |
> > | LARGE | SCENT (w/o C) | 430 ±295 | 1817 ±1219 | 0.58 ±0.01 | 2454 ±637 |
> > | LARGE | SCENT (C)  | 116 ±44 | 3165 ±2437 | 0.67 ±0.02 | 1213 ±186 |
> > | LARGE | SCENT (D) | 112 ± 129 | 502 ± 348 | 0.57 ± 0.02 | - |
> > | LARGE | SCENT (P) | 99.0 ± 93.5 | 536 ± 298 | 0.57 ± 0.03 | 2531 ± 563 |

---

> > > ### Comment · Reviewer_E4oU · 2025-08-07
> > >
> > > Thank you for the detailed and thoughtful response. I appreciate the effort you put into addressing the points I raised. Most of my concerns have been resolved.
> > >
> > > Given the overall quality of the work and the improvements made in response to the review, I will maintain my positive rating.

---

### Official Review · Reviewer_ptir · 2025-07-19

**Clarity:** 2
**Significance:** 3
**Originality:** 2
**Rating:** 3
**Confidence:** 3

**Summary:**

In the manuscript, the authors present SCENT, an approach to template-based molecule generation. SCENT aims to minimize synthesis cost and increase molecule diversity. The authors propose recursive cost guidance to approximate synthesis cost, and design a dynamic library mechanism to reuse the intermediates with high reward. The evaluation results suggest the advantages of SCENT over the existing approaches.

**Questions:**

I strongly recommend the authors to show some case studies, including the yielded molecules, the reactions that yield them, together with synthesis costs. The ground-truth synthesis costs are also expected to judge the performance of the designed Recursive Cost Guidance module.

**Ethical Concerns:**

["NO or VERY MINOR ethics concerns only"]

**Limitations:**

Yes

**Paper Formatting Concerns:**

Line 79-80: a sentence contains two “which” clauses
Line 80: the term “MDP” is used without definition.
Eq 4 and 8: $l$ was not defined.
Eq 10: $l_i$ was not used in the definition of $P_B^D$
Fig. 1: The summary sentence is missing in the caption.

**Quality:**

3

**Strengths And Weaknesses:**

Strength:
SCENT estimates synthesis cost using a machine learning module, thereby enabling it to identify reaction pathways with low cost.

Weakness:
What is the ground-truth synthesis cost of a molecule? An example is expected to state the performance of the prediction of the synthesis cost. It is also much easier to distinguish the advantages of the presented approach if wet-lab experimental results are available.

---

> ### Author Rebuttal · Authors · 2025-07-31
>
> We thank the reviewer for their feedback. We have discussed all the issues raised and incorporated the suggestion to include case studies on generated synthesis trajectories which strengthen our paper. Given these revisions, we kindly hope the reviewer will consider updating their score.
>
> ## Ground-truth synthesis cost of a molecule
>
> As SCENT is designed for *de novo* molecular discovery, most of its outputs are novel and have not been synthesized before. Consequently, empirical synthesis costs are not available. The only definitive way to validate these costs would be through wet-lab experiments, which we agree would be valuable, but fall outside the scope of this work.
>
> ## Wet-lab experiments
>
> We appreciate the reviewer’s concern regarding the absence of wet-lab validation, which we fully acknowledge would significantly strengthen the benchmark. However, we believe this falls outside the typical scope of a machine learning conference. Our goal is to introduce and evaluate SCENT as a computational framework; while experimental validation is important, we argue it is not essential at this stage for demonstrating the method’s value in algorithmic and modeling terms. We see wet-lab validation as an important next step and a promising direction for future collaboration with experimental groups.
>
> ## Case studies
>
> We appreciate the reviewer’s suggestion and have conducted a detailed case study analyzing synthesis trajectories proposed by our model. While we are unable to include any images at this rebuttal stage, specific examples, including the generated molecules, the reactions used to construct them, and the associated synthesis costs, are provided in the next *Pathway Examples* section. The detailed analysis of the pathways can be found in our response to Reviewer **9Lb3**, under the section *Pathways Analysis*. It demonstrates that the Recursive Cost Guidance module effectively reduces synthesis cost by deferring the use of expensive fragments until later stages in the trajectory and favoring cheaper fragment’s alternatives when available. We will include the entire analysis, along with visualizations in the revised paper.
>
> ## Pathways Examples
> The extensive analysis of those synthesis pathways can be found in response to Reviewer **9Lb3**, under the section *Pathways Analysis*.
>
> ### **Example 1, SMALL settings.**
>
> -------------------
>
> **SCENT (w/o C)**
>
> Step 1:
>
> OB(c(cc1)cc2c1[nH]cc2)O.Nc3c(O)cc(Br)cc3>>Nc4c(O)cc(c(cc5)cc6c5[nH]cc6)cc4
>
> Reactant_1 cost: 12.56
>
> Reactant_2 cost: 0.43
>
> Reaction Yield: 75%
>
> Product cost: 17.32
>
> Step 2:
>
> Nc1c(O)cc(c(cc2)cc3c2[nH]cc3)cc1.O=C(C4CNC4)O>>c5(cc(c(cc6)cc7c6nc(C8CNC8)o7)cc9)c9[nH]cc5
>
> Reactant_1 cost: 17.32
>
> Reactant_2 cost: 0.19
>
> Reaction Yield: 70%
>
> Product cost: 25.01
>
> Step 3:
>
> ClCC1CC1.c2(cc(c(cc3)cc4c3nc(C5CNC5)o4)cc6)c6[nH]cc2>>c78n(CC9CC9)ccc7cc(c(cc%10)cc%11c%10nc(C%12CNC%12)o%11)cc8
>
> Reactant_1 cost: 0.33
>
> Reactant_2 cost: 25.01
>
> Reaction Yield: 75%
>
> Product cost: 33.80
>
> Step 4:
>
> O=C(C1CNC1)O.c23n(CC4CC4)ccc2cc(c(cc5)cc6c5nc(C7CNC7)o6)cc3>>O=C(C8CCC8)N(C9)CC9c%10oc%11cc(c%12ccc%13n(CC%14CC%14)ccc%13c%12)ccc%11n%10
>
> Reactant_1 cost: 0.19
>
> Reactant_2 cost: 33.80
>
> Reaction Yield: 75%
>
> **Product cost: 45.32**
>
> -------------------
>
> **SCENT (C)**
>
> Step 1:
>
> O=C(C1CNC1)O.Nc2c(O)cc(Br)cc2>>Brc(cc3)cc4c3nc(C5CNC5)o4
>
> Reactant_1 cost: 0.19
>
> Reactant_2 cost: 0.43
>
> Reaction Yield: 70%
>
> Product cost: 0.89
>
> Step 2:
>
> Brc(cc1)cc2c1nc(C3CNC3)o2.OB(c(cc4)cc5c4[nH]cc5)O>>c6(cc(c(cc7)cc8c7nc(C9CNC9)o8)cc%10)c%10[nH]cc6
>
> Reactant_1 cost: 0.89
>
> Reactant_2 cost: 12.56
>
> Reaction Yield: 75%
>
> Product cost: 17.93
>
> Step 3:
>
> ClCC1CC1.c2(cc(c(cc3)cc4c3nc(C5CNC5)o4)cc6)c6[nH]cc2>>c78n(CC9CC9)ccc7cc(c(cc%10)cc%11c%10nc(C%12CNC%12)o%11)cc8
>
> Reactant_1 cost: 0.33
>
> Reactant_2 cost: 17.93
>
> Reaction Yield: 75%
>
> Product cost: 24.35
>
> Step 4:
>
> O=C(C1CNC1)O.c23n(CC4CC4)ccc2cc(c(cc5)cc6c5nc(C7CNC7)o6)cc3>>O=C(C8CCC8)N(C9)CC9c%10oc%11cc(c%12ccc%13n(CC%14CC%14)ccc%13c%12)ccc%11n%10
>
> Reactant_1 cost: 0.19
>
> Reactant_2 cost: 24.35
>
> Reaction Yield: 75%
>
> **Product cost: 32.72**
>
> -------------------
>
> **SCENT (C+D)**
>
> Step 1:
>
> Reactant_2 (c2(cc(c(cc3)cc4c3nc(C5CNC5)o4)cc6)c6[nH]cc2) was taken from the Dynamic Library
>
> ClCC1CC1.c2(cc(c(cc3)cc4c3nc(C5CNC5)o4)cc6)c6[nH]cc2>>c78n(CC9CC9)ccc7cc(c(cc%10)cc%11c%10nc(C%12CNC%12)o%11)cc8
>
> Reactant_1 cost: 0.33
>
> Reactant_2 cost: 17.93
>
> Reaction Yield: 75%
>
> Product cost: 24.35
>
> Step 2:
>
> O=C(C1CNC1)O.c23n(CC4CC4)ccc2cc(c(cc5)cc6c5nc(C7CNC7)o6)cc3>>O=C(C8CCC8)N(C9)CC9c%10oc%11cc(c%12ccc%13n(CC%14CC%14)ccc%13c%12)ccc%11n%10
>
> Reactant_1 cost: 0.19
>
> Reactant_2 cost: 24.35
>
> Reaction Yield: 75%
>
> **Product cost: 32.72**
>
> -------------------
>
> ### **Example 2, SMALL settings.**
>
> -------------------
>
> **SCENT (w/o C)**
>
> Step 1:
>
> O=C([C@H]1CCCNC1)O.O=C([C@H]2CCCN2)O>>O=C([C@H]3CCCN3C([C@H]4CCCNC4)=O)O
>
> Reactant_1 cost: 19.37
>
> Reactant_2 cost: 0.10
>
> Reaction Yield: 75%
>
> Product cost: 25.97
>
> Step 2:
>
> O=C([C@H]1CCCN1C([C@H]2CCCNC2)=O)O.NCc(c[nH]3)c4c3cccc4>>O=C([C@H]5CCCN5C([C@H]6CCCN(C(NCc(c[nH]7)c8c7cccc8)=O)C6)=O)O
>
> Reactant_1 cost: 25.97
>
> Reactant_2 cost: 16.81
>
> Reaction Yield: 75%
>
> Product cost: 57.04
>
> Step 3:
>
> O=C([C@H]1CCCN1C([C@H]2CCCN(C(NCc(c[nH]3)c4c3cccc4)=O)C2)=O)O.NC5CC5>>O=C([C@H]6CCCN6C([C@H]7CCCN(C(NCc(c[nH]8)c9c8cccc9)=O)C7)=O)NC%10CC%10
>
> Reactant_1 cost: 57.04
>
> Reactant_2 cost: 0.02
>
> Reaction Yield: 75%
>
> Product cost: 76.07
>
> Step 4:
>
> O=C(NC1CC1)[C@H]2CCCN2C([C@H]3CCCN(C3)C(NCc4c5ccccc5[nH]c4)=O)=O.ClCC6CC6>>O=C(NC7CC7)[C@H]8CCCN8C([C@H]9CCCN(C9)C(NCc%10c%11ccccc%11n(CC%12CC%12)c%10)=O)=O
>
> Reactant_1 cost: 76.07
>
> Reactant_2 cost: 0.33
>
> Reaction Yield: 75%
>
> **Product cost: 101.87**
>
> -------------------
>
> **SCENT (C)**
>
> Step 1:
>
> NC1CC1.O=C([C@H]2CCCN2)O>>O=C([C@H]3CCCN3)NC4CC4
>
> Reactant_1 cost: 0.02
>
> Reactant_2 cost: 0.10
>
> Reaction Yield: 75%
>
> Product cost: 0.16
>
> Step 2:
>
> O=C([C@H]1CCCN1)NC2CC2.O=C([C@H]3CCCNC3)O>>O=C([C@H]4CCCN4C([C@H]5CCCNC5)=O)NC6CC6
>
> Reactant_1 cost: 0.16
>
> Reactant_2 cost: 19.37
>
> Reaction Yield: 75%
>
> Product cost: 26.04
>
> Step 3:
>
> O=C([C@H]1CCCN1C([C@H]2CCCNC2)=O)NC3CC3.NCc(c[nH]4)c5c4cccc5>>O=C([C@H]6CCCN6C([C@H]7CCCN(C(NCc(c[nH]8)c9c8cccc9)=O)C7)=O)NC%10CC%10
>
> Reactant_1 cost: 26.04
>
> Reactant_2 cost: 16.81
>
> Reaction Yield: 75%
>
> Product cost: 57.14
>
> Step 4:
>
> O=C(NC1CC1)[C@H]2CCCN2C([C@H]3CCCN(C3)C(NCc4c5ccccc5[nH]c4)=O)=O.ClCC6CC6>>O=C(NC7CC7)[C@H]8CCCN8C([C@H]9CCCN(C9)C(NCc%10c%11ccccc%11n(CC%12CC%12)c%10)=O)=O
>
> Reactant_1 cost: 57.14
>
> Reactant_2 cost: 0.33
>
> Reaction Yield: 75%
>
> **Product cost: 76.63**
>
> -------------------
>
> **SCENT (C+D)**
>
> Step 1:
>
> NC1CC1.O=C([C@H]2CCCN2)O>>O=C([C@H]3CCCN3)NC4CC4
>
> Reactant_1 cost: 0.02
>
> Reactant_2 cost: 0.10
>
> Reaction Yield: 75%
>
> Product cost: 0.16
>
> Step 2:
>
> O=C([C@H]1CCCN1)NC2CC2.O=C([C@H]3CCCNC3)O>>O=C([C@H]4CCCN4C([C@H]5CCCNC5)=O)NC6CC6
>
> Reactant_1 cost: 0.16
>
> Reactant_2 cost: 19.37
>
> Reaction Yield: 75%
>
> Product cost: 26.04
>
> Step 3:
>
> Reactant_2 (NCc1c2ccccc2n(CC3CC3)c1) was taken from the Dynamic Library
>
> O=C([C@H]1CCCN1C([C@H]2CCCNC2)=O)NC3CC3.NCc4c5ccccc5n(CC6CC6)c4>>O=C(NC7CC7)[C@H]8CCCN8C([C@H]9CCCN(C9)C(NCc%10c%11ccccc%11n(CC%12CC%12)c%10)=O)=O
>
> Reactant_1 cost: 26.04
>
> Reactant_2 cost: 22.86
>
> Reaction Yield: 75%
>
> **Product cost: 65.20**
>
> -------------------
>
> ### **Example 3, MEDIUM settings.**
>
> -------------------
>
> **SCENT (w/o C)**
>
> Step 1:
>
> N#CCCNc1cc(OCCO2)c2cc1.Cn3cc(CNc4cc(C(O)=O)ccc4)cc3>>Cn5cc(CNc6cc(C(N(c7cc(OCCO8)c8cc7)CCC#N)=O)ccc6)cc5
>
> Reactant_1 cost: 125.31
>
> Reactant_2 cost: 279.35
>
> Reaction Yield: 75%
>
> **Product cost: 539.55**
>
> -------------------
>
> **SCENT (C)**
>
> Step 1:
>
> N#CCCNc1cc(OCCO2)c2cc1.Nc3cc(c4cc(C(O)=O)ccc4)ccc3>>N#CCCN(C(c5cc(c6cc(N)ccc6)ccc5)=O)c7cc(OCCO8)c8cc7
>
> Reactant_1 cost: 125.31
>
> Reactant_2 cost: 27.27
>
> Reaction Yield: 75%
>
> **Product cost: 203.45**

---

> > ### Comment · Reviewer_ptir · 2025-08-08
> >
> > I  appreciate the author's efforts to address my concerns.
> >
> > I also agree with the authors on the point that wet-lab experiments fall out of the scope of a paper focusing on algorithms.
> >
> > I am glad to see a case study. Could the authors add more sentences to explain the calculation of "product cost"?

---

> > > ### Author Response · Authors · 2025-08-08
> > >
> > > We thank the reviewer for their feedback on our rebuttal. The "product cost" represents the estimated cost of producing 1 nmol of the target compound, calculated based on reaction yield and the cost of the reactants. Specifically, it is defined as:
> > > (cost of reactant_1 + cost of reactant_2) / reaction yield. Reactant_1 from step $n$ corresponds to the product from step $n–1$, except in the first step, where both reactants are selected from the building block library. We will revise the figure captions in the manuscript to explicitly include this explanation. We appreciate the reviewer’s suggestion to improve clarity on this point.

---

### Official Review · Reviewer_QKfB · 2025-07-20

**Clarity:** 2
**Significance:** 3
**Originality:** 3
**Rating:** 4
**Confidence:** 4

**Summary:**

This paper proposes SCENT (Scalable and Cost-Efficient de Novo Template-Based Molecular Generation), which combines several strategies with Reaction GFlowNet to generate diverse molecules while reducing synthesis cost. These strategies include Recursive Cost Guidance, which enables the model to focus on efficient sampling; Decomposability Guidance, which contributes to overall performance; an Exploitation Penalty; and a Dynamic Library of building blocks to increase mode coverage. In this way, the proposed method achieves state-of-the-art results among template-based approaches.

**Questions:**

**"Regarding the effectiveness of the Dynamic Library (D):**

The results presented in Tables 1–4 indicate that the performance improvement from adding the Dynamic Library (i.e., comparing "SCENT (C)" to "SCENT (C+D)") varies significantly across different tasks and settings. For instance, while it yields notable gains in some configurations (e.g., GSK3β, LARGE setting in Table 3), its contribution appears more modest in others (e.g., sEH proxy under the MEDIUM setting in Tables 1 and 2, and JNK3 in Table 4).

Since the Dynamic Library is intended to expand the explorable space by enabling richer, tree-structured synthesis pathways, one might anticipate more consistent performance gains.

- Could the authors provide an analysis or hypothesis for this variability?
- What factors might influence when the Dynamic Library is most effective versus when its impact is limited?

**Regarding the performance of the Recursive Cost Guidance surrogate model:**

The effectiveness of the Recursive Cost Guidance framework critically depends on the predictive accuracy of the surrogate model, $\hat{c}_B$. However, the paper does not report direct performance metrics for this model (e.g., validation loss or correlation with the target synthesis costs).

- Could the authors provide more detail on the performance of this surrogate model?
- Additionally, what are the potential consequences if the surrogate model makes inaccurate predictions? How robust is the overall method to such prediction errors?

**Ethical Concerns:**

["NO or VERY MINOR ethics concerns only"]

**Final Justification:**

Considering the overall evaluation and the discussion so far, I have decided to maintain my rating.

**Limitations:**

As noted in the Weaknesses section, the authors do not address the scalability of their approach with respect to the number of reaction templates. This may constitute a potential limitation, particularly as the size of the action space increases in realistic scenarios.

**Paper Formatting Concerns:**

No paper formatting concerns.

**Quality:**

3

**Strengths And Weaknesses:**

### **Strengths**

This paper introduces SCENT, a GFlowNet-based framework aimed at generating molecules that are both synthetically accessible and cost-efficient. Its key contributions include:

- **Practical Relevance:** The study addresses an important challenge in drug discovery—producing molecules that are not only satisfied with targets but also straightforward and inexpensive to synthesize. By introducing an estimated synthesis cost into the generative process, the work moves closer to real-world applicability than approaches that focus solely on property optimization.
- **Methodological Contributions (Originality & Quality):**
    - The proposed **Recursive Cost Guidance** framework provides a systematic way to steer GFlowNet’s backward policy. Approximating an intractable recursive cost with a learned surrogate model ($\hat{c}^B$) improves efficiency and not only directs generation toward lower-cost synthesis pathways but also increases mode coverage especially in settings with large building block libraries.
    - To mitigate the over-exploitation observed with Recursive Cost Guidance, the authors introduce and show a simple yet effective exploitation-penalty term ($P_{E}$), as illustrated in Figure 5 and quantified in Table 1 for the medium- and large-building-block settings.
    - The **Dynamic Library** (D) mechanism offers a form of augmenting in molecular generation by caching and reusing high-reward intermediates, enabling the construction of richer, tree-structured synthesis pathways shown in such as Table 1 and 3 of Large settings.
- **Empirical Evaluation (Quality):**
    - Experiments cover three library sizes (SMALL, MEDIUM, LARGE) and three molecular design tasks, providing evidence of robustness and scalability.
    - SCENT outperforms several template-based GFlowNets (RGFN, SynFlowNet, RxnFlow) on metrics such as the number of high-reward compounds, modes, scaffolds discovered, and cost.
    - Ablation studies isolate the effects of Cost Guidance, Dynamic Library, and Exploitation Penalty, clarifying each component’s contribution.

### **Weaknesses**

Several aspects remain insufficiently documented:

- **Quantitative Computational Cost Analysis (Quality):** The paper does not report key compute metrics (e.g., GPU hours or other resource usage) for the small-, medium-, and large-setting configurations, nor does it compare them to other methods. The NeurIPS checklist notes that these data were not tracked, leaving efficiency claims unverified.
- **Scalability with Reaction Templates (Quality):** While the study examines scaling with the number of *building blocks*, it does not evaluate how performance changes as the number of **reaction templates** grows, which would enlarge the action space and pose additional learning challenges.
- **Lack of a Limitations Discussion (Clarity & Quality):** The paper omits a dedicated section on limitations. Issues such as the simplified cost model (which excludes solvents and purification) or potential failure modes of the guidance model are therefore not discussed, reducing transparency.
- **Variable Performance (Quality):** In the JNK3 task (Table 4, SMALL and MEDIUM settings), the baseline RGFN discovers more diverse, high-reward modes and scaffolds than SCENT. Although the authors note that SCENT (C+P) generates the cheapest molecules, the paper provides limited analysis of why SCENT underperforms in this setting, which would help clarify the method’s boundaries.
- **Hyper-parameter Settings (Minor):** Each component involves multiple hyper-parameters, yet the paper does not analyze how sensitive performance is to their values. Because an explore–exploit trade-off is plausible when increasing the number of building blocks and reaction templates, commenting on the method’s robustness under hyper-parameter variation would strengthen the work.

---

> ### Author Rebuttal · Authors · 2025-07-31
>
> We truly value the reviewer’s detailed feedback and thoughtful recommendations. We have integrated all the suggestions, with the GPU resource metrics, cost model performance analysis, and the hyperparameter sensitivity study significantly enhancing the quality of our paper. We have addressed all the concerns raised and ask the reviewer to take these revisions into account during their evaluation.
>
> ## Computational resources
>
> As requested by the reviewers, we report GPU runtimes (in minutes) for all baselines on the sEH proxy, averaged over 3 random seeds using a V100 32GB GPU. All methods used a batch size of 64, except SynFlowNet, which required batch size 32 on the LARGE setting even after optimizing its policy code to fit within memory limits.
>
> | Model | SMALL | MEDIUM | LARGE |
> | --- | --- | --- | --- |
> | RGFN | 2882 ± 115 | 3892 ± 153 | 4455 ± 392 |
> | SynFlowNet | 200 ± 3 | 2350 ± 162 | 8416 ± 19 |
> | RxnFlow | 891 ± 4 | 3304 ± 44 | 3399 ± 50 |
> | SCENT (C+D+P) | 800 ± 15 | 2849 ± 144 | 3688 ± 317 |
>
> RGFN shares a part of the codebase with SCENT, allowing for direct comparison. We observe consistent improvements across all settings, particularly in the SMALL configuration. While SynFlowNet is faster on SMALL and MEDIUM, it struggles on LARGE, even with our memory optimizations. However, differences in codebases make direct comparisons less definitive.
>
> Note that our implementation is not optimized for runtime and could benefit from further improvements.
>
> ## Scaling with number of reactions
>
> We agree that scaling reaction-based generative models with the number of reaction templates is an important factor, omitted from the paper. However, curating a large, high-quality reaction set is a resource-intensive process that cannot be completed within the rebuttal period. Even with automated extraction (which would reduce reaction quality) the computational demands of evaluating all baselines exceed what is feasible in one week. We believe that our experiments already demonstrate the merits of our approach. While scaling with more templates could add value, it would not alter the key conclusions of our work - we propose this as a promising direction for future research.
>
> ## Lack of limitation section
>
> We apologize for the lack of limitation section, we are going to include the following limitation summary in the revised paper:
>
> “While our study demonstrates the effectiveness of SCENT in generating cost-efficient and synthesizable molecules, several limitations remain.
>
> First, the synthesis cost estimates are simplified and heuristic, omitting factors such as solvent usage, purification steps, hazardous reagents, reaction conditions, and scalability. Consequently, the current cost model should be regarded as a first-order approximation. More accurate estimation would require lab-specific calibration, including robust yield prediction models and vendor-specific building block pricing.
>
> SCENT’s performance varies across different biological targets, indicating that its effectiveness may depend on task-specific factors. Identifying and addressing these bottlenecks is an important direction for future work. Key areas of improvement include refining the exploitation behavior of the Dynamic Library, and enhancing the generalizability of the cost predictor model used in Synthesis Cost Guidance. Additionally, we do not assess performance as the number of reaction templates increases, an important scalability consideration, as this expansion significantly enlarges the action space and introduces additional learning challenges.
>
> Our framework emphasizes synthesizability and cost, but does not incorporate toxicity or broader ADMET properties, which are critical for downstream drug development. Finally, we acknowledge the absence of wet-lab validation, which remains the definitive benchmark for evaluating the practical utility of our approach. We leave this to future work.”
>
> ## Why SCENT underperforms on JNK3?
>
> Thank you for the thoughtful feedback. One notable characteristic of the JNK3 task is its sparse reward landscape - the number of discovered modes is significantly lower than in sEH and GSK3β. We hypothesize that in such settings, excessive exploitation driven by strong cost guidance may hinder mode discovery. Reducing the Synthesis Cost Guidance temperature (\$\alpha\_S\$) could help balance exploration and exploitation, potentially improving diversity at some cost trade-off. SCENT was not specifically tuned for JNK3, and experiments to test this hypothesis are ongoing - we will share the results as soon as they are available.
>
> ## Hyperparameters
>
> To assess SCENT (C+D+P) sensitivity to hyperparameters, we performed an ablation study on sEH MEDIUM, varying each hyperparameter around the paper’s chosen values (bolded in tables) over 3k iterations. Results indicate SCENT is generally robust, with no catastrophic performance drops. However, some hyperparameters notably affect outcomes; for example, simple tuning of $\alpha_D$ can boost discovered modes by ~20%. This highlights potential for further improvements through targeted tuning, left for future work. Detailed results follow.
>
> | model | modes > 8.0 | scaff. > 8.0 | reward  | cost > 8.0 |
> | --- | --- | --- | --- | --- |
> | Synthesis Cost Guidance (temperature $\alpha_S$) | - | - | - | - |
> | $\alpha_S=2$ | 16036 ± 1153 | 22292 ± 1652 | 7.55 ± 0.05 | 1300 ± 82 |
> | **$\alpha_S=5$** | 16373 ± 1827 | 39721 ± 3574 | 7.61 ± 0.03 | 1163 ± 81 |
> | $\alpha_S=10$ | 13566 ± 4207 | 73916 ± 2516 | 7.87 ± 0.04 | 898 ± 42 |
> | Decomposability Guidance (temperature  $\alpha_D$) | - | - | - | - |
> |  $\alpha_D=2$ | 20023 ± 1484 | 45483 ± 2490 | 7.74 ± 0.04 | 1137 ± 130 |
> |  $**\alpha_D=5$** | 16373 ± 1827 | 39721 ± 3574 | 7.61 ± 0.03 | 1163 ± 81 |
> |  $\alpha_D=10$ | 20444 ± 2055 | 44938 ± 3111 | 7.74 ± 0.08 | 1141 ± 80 |
> | Dynamic Library (number of added molecules) | - | - | - | - |
> | L=200 | 19700 ± 1919 | 49119 ± 6425 | 7.72 ± 0.05 | 1087 ± 183 |
> | **L=400** | 16373 ± 1827 | 39721 ± 3574 | 7.61 ± 0.03 | 1163 ± 81 |
> | L=800 | 18373 ± 987 | 48016 ± 4302 | 7.71 ± 0.08 | 1060 ± 66 |
> | Dynamic Library (frequency of updates) | - | - | - | - |
> | T=500 | 15670 ± 2482 | 39701 ± 3676 | 7.66 ± 0.12 | 1130 ± 106 |
> | **T=1000** | 16373 ± 1827 | 39721 ± 3574 | 7.61 ± 0.03 | 1163 ± 81 |
> | T=2000 | 16148 ± 1753 | 39099 ± 3341 | 7.66 ± 0.08 | 1141 ± 82 |
> | Exploitation Penalty (initial $\gamma_0$) | - | - | - | - |
> | $\gamma_0=0.5$ | 20690 ± 1512 | 47220 ± 3676 | 7.69 ± 0.06 | 1154 ± 48 |
> | **$\gamma_0=1.0$** | 16373 ± 1827 | 39721 ± 3574 | 7.61 ± 0.03 | 1163 ± 81 |
> | $\gamma_0=2.0$ | 20655 ± 1514 | 45990 ± 1593 | 7.75 ± 0.04 | 1111 ± 77 |
> | Exploitation Penalty ($\Delta \gamma$) | - | - | - | - |
> | $\Delta \gamma=0.0$ | 20025 ± 762 | 43432 ± 796 | 7.75 ± 0.02 | 1222 ± 65 |
> | **$\Delta \gamma=0.2$** | 16373 ± 1827 | 39721 ± 3574 | 7.61 ± 0.03 | 1163 ± 81 |
> | $\Delta \gamma=0.4$ | 17189 ± 2234 | 42718 ± 2799 | 7.69 ± 0.04 | 1108 ± 165 |
>
> ## Why Dynamic Library underperforms in some settings?
>
> While it is true that the dynamic library sometimes shows lower mode discovery in the online setting, it consistently yields higher-reward samples on average during inference. This behavior arises because dynamic library updates can occasionally favor fragments that bias exploration toward a narrower region of the search space. While this region may be high-reward, this comes at the cost of reduced diversity, which explains the lower mode discovery numbers highlighted in the review.
>
> This trade-off suggests that the main limiting factor is the fragment selection criterion. Carefully designing this criterion to maintain diversity while still encouraging promising fragments is key to improving exploration and fully realizing the benefits of the dynamic library.
>
> ## Performance of cost models
>
> Thank you for the helpful suggestion regarding the cost prediction model. To better understand its performance, we trained it separately on a subset of molecules from SCENT trajectories in the SMALL (C) setting. For validation, we used a held-out set from the same setting and computed ground-truth costs using the full recurrence from Equation 7, whereas the training set relied on approximate estimates to simulate the online learning scenario.
>
> We observed steady improvement during training, with validation R² peaking at 0.77 before overfitting set in, eventually declining to ~0.6, while training R² continued to rise to 1.0. While we’re unable to include plots at this rebuttal stage, we’ve added them to the revised version of the paper.
>
> It’s worth noting that in the online setting, the model is trained on trajectories sampled by SCENT itself, helping it learn to identify low-cost backward trajectories. However, during inference, SCENT only uses forward sampling and does not rely on the cost predictor. Thus, while the model’s validation performance may influence trajectory selection during training, it does not affect SCENT’s inference procedure directly.
>
> ## Consequences of inaccurate prediction model
>
> The consequences of inaccurate cost predictions are assessed in Appendix G.2.2. We compared our model with a low-quality cost estimator that outputs a constant value for every input molecule, where this constant is the mean cost of all the building blocks in the given setting. It is important to note that we only modify the recursive component of the $C^S$ function from Equation (5), keeping the fragment costs unchanged. As shown in Table 8, this constant estimator can reduce costs and boost SCENT’s performance in MEDIUM and LARGE settings, but the machine learning-based cost model used in the main paper delivers far greater improvements, especially in cost reduction. This confirms that cost prediction quality is crucial for SCENT’s

---

> > ### Comment · Reviewer_QKfB · 2025-08-06
> >
> > Thank you for your detailed response and for conducting the additional experiments to address my questions.
> >
> > While I still have some minor reservations about the scalability with respect to the number of reaction templates, I acknowledge that a full analysis would be difficult during the rebuttal period. Nevertheless, the results from the additional experiments you provided demonstrate that the proposed method is generally robust and effective, and remains stable under hyperparameter variations.
> >
> > Regarding the JNK3 task, I appreciate your clarification. While my underlying question about the definitive cause for the limited diversity gains is not fully resolved, I agree that your point is well-taken. The ability of SCENT to achieve performance comparable to RGFN while maintaining a lower synthesis cost is a strong demonstration of the method's robustness and practical utility.
> >
> > With these clarifications, my major concerns have been adequately addressed.

---

> > > ### Author Response · Authors · 2025-08-07
> > >
> > > We appreciate your understanding of the limited resource budget during the rebuttal phase. We fully agree on the importance of scaling with respect to the number of reactions and share the interest in identifying the underlying causes of the limited diversity gains observed on certain tasks, which turned out to be more complex than we initially anticipated. Both points are now explicitly acknowledged in the revised Limitations section.
> > >
> > > Finally, we thank you for the positive assessment of our rebuttal.

---

> ### Author Response · Authors · 2025-08-04
> **Why SCENT underperforms on JNK3?**
>
> Following up on our earlier hypothesis regarding SCENT’s underperformance on JNK3, we conducted additional experiments to test whether reducing the Synthesis Cost Guidance temperature ($\alpha_S$) could improve mode discovery. Specifically, we ran SCENT (C + D + P) with $\alpha_S=2$ instead of the default $\alpha_S=5$ on all settings.
>
> The results did not support the hypothesis: lowering the temperature increased synthesis cost but did not consistently improve diversity.
>
> The results clearly refute the hypothesis: lowering the temperature led to an increase in synthesis cost but did not produce consistent improvements in diversity.
>
> While we do not have a definitive explanation for the limited diversity gains on JNK3, it is important to emphasize that SCENT’s performance remains robust. The number of discovered modes on JNK3 is comparable to RGFN, yet SCENT achieves this with a lower synthesis cost, demonstrating its efficiency even in challenging sparse-reward environments. We will incorporate these findings into the final version of the paper.

---

> > ### Author Response · Authors · 2025-08-05
> > **Performance of decomposability models**
> >
> > Following up on the performance of cost models, we conducted the same analysis but for decomposability prediction that is core to speeding up SCENT.
> >
> > Same as the cost models, we observed steady improvement during training, with validation accuracy peaking at 0.97 and a F1 Score of 0.93, while training accuracy and F1 Score both rise to 1.0. We will include the corresponding plots in the revised manuscript.

---

### Official Review · Reviewer_9Lb3 · 2025-07-21

**Clarity:** 4
**Significance:** 3
**Originality:** 3
**Rating:** 4
**Confidence:** 4

**Summary:**

The authors introduce SCENT, a GFlowNet (GFN)-based framework designed to overcome limitations of prior template-based approaches for synthesizable molecular generation: (1) not considering "synthesis cost" (2) weak scalability when dealing with large building-block libraries and (3) poor/inefficient use of small fragments. The paper proposes to tackle these problems by introducing "Recursive Cost Guidance", where a learned cost predictor $\hat c_{B}^{S}$ steers the backward policy toward low-cost intermediates (defined by building block stock-prices and reaction yields) and a decomposability model that prevents invalid retrosynthetic steps. In addition, the authors propose a forward-policy regularizer that penalizes repeated state-action visits to counteract exploitation introduced by the cost guidance ("Exploitation Penalty"). Lastly, the authors explore a fragment pool augmentation method where they periodically add high-reward intermediates to the pool, ranked by their expected utility.

**Questions:**

* What happens if the stock-prices of the building blocks change? Do you need to retrain all modules from scratch, or is the system robust enough to just update the cost predictor and fine-tune? Brief analysis/experiments regarding "warm-start" scenario could add value to the draft and its applicability to real-world scenarios.
* It’s unclear how the “Dynamic Building Block Library” truly expands the space of synthesizable molecules (mentioned in line 198). In principle, any intermediate added to the library was already reachable via longer retrosynthetic trajectories, so the *theoretical* reachable set should remain unchanged. Could the authors clarify what they mean by “expanding” the space versus merely compressing trajectory length?
* How sensitive is the proposed approach to the Dynamic Library hyperparameters, specifically, the frequency of pool updates $T$ and the number of added intermediates $L$?
* How are the reaction yield estimates computed for the cost predictor? Are they the average of all available experimental yields per template, the highest reported yield, or some other summary statistic? Clarifying this would help assess the bias of the cost-guidance module.

**Ethical Concerns:**

["NO or VERY MINOR ethics concerns only"]

**Final Justification:**

Considering the rebuttal responses, I maintain my positive recommendation. The paper makes a meaningful contribution to the literature on synthesizable molecular generation by directly integrating cost considerations into the GFlowNet architecture (guidance), rather than treating them solely as a reward signal. The authors’ warm-start study shows robustness to changes in building-block costs through a post-training process, which strengthens the practical applicability of the approach.

Their clarifications and additional experiments on the Dynamic Library’s behavior, hyperparameter sensitivity, and qualitative pathway analysis addressed the key concerns I raised. While certain limitations remain (e.g., overly simplified yield and cost models), these seem to be reasonable given the scope of the work. Overall, the manuscript is well-written, technically sound, and advances aspects of synthesizable generation methods, particularly in linking cost efficiency with route generation.

**Limitations:**

The authors do not address the limitations of their work in the draft.

- **Unrealistic “one-step” fragments.** The Dynamic Library treats discovered intermediates as if they were freely available building blocks, ignoring the multistep chemistry (and costs) needed to actually make them before use. This should probably be included in the expected utility in some form.

**Paper Formatting Concerns:**

* Typo in line 204: "This procedure is update up to $N_{add}$ times."

**Quality:**

3

**Strengths And Weaknesses:**

**Strengths:**
- The paper is well-written and logically structured, which makes it easy to follow.
- The authors clearly identify and address key drawbacks of existing template-based GFN approaches for synthesizable molecular generation.
- Comprehensive ablation study (Table 1, Sections 4.3–4.5, Appendix) help quatify the individual contributions of each proposed component.
- **Recursive Cost Guidance** directly integrates synthetic cost considerations into the backward policy rather than treating them solely as a reward signal, resulting in consistent reductions in average building-block price and shorter synthesis paths. This is important since rather than aiming to match the full support of high and low-cost routes/yields, this approach focuses the backward policy on the most synthetically viable, high-reward modes, which aligns directly with the practical objective of discovering cost-efficient molecules.
- **Exploitation Penalty** effectively counteracts the over-exploitation introduced by cost guidance, as demonstrated by reduced scaffold-revisit rates in Figure 5.

**Weaknesses**
- Yields are considered to be constant for each template, when these are known to show high variance across reactants, reagents, solvents and conditions.
- Cost of a product only considers building blocks, ignoring solvents, catalysts or reagents. As a proxy, these values could be integrated in some sort of cost for the templates themselves.
- No limitations section or future work.
- No further qualitative analysis on the quality of the generated structures and their corresponding routes. Is there any noticeable introduced bias on the generated routes/structures after adding each of the proposed components?
- Building-block stock prices fluctuate and depend on purchase scale and vendor. The paper does not explore how often price updates require warm-starting or full retraining, nor quantify the impact of price uncertainty on model outputs.

---

> ### Author Rebuttal · Authors · 2025-07-31
>
> We sincerely appreciate the reviewer’s thoroughness and insightful suggestions. We have incorporated all of them and believe that the warm-start study and qualitative analysis of the pathways, in particular, have significantly strengthened our paper. Regarding the concerns raised, we have carefully addressed each one and kindly encourage the reviewer to consider this in their evaluation.
>
> ## Warm-start study
>
> We found the “warm-start” scenario compelling and conducted an experiment on SCENT (C+D+P) to evaluate system robustness to changes in fragment costs.
>
> At iteration 3000, we randomly reshuffled fragment costs and continued training without resetting model parameters, only updating the costs estimates used to train the cost predictor. As expected, the trajectory cost spiked immediately after the change but quickly decreased, converging to the same level reached by a model trained from scratch with the new cost assignments (within ~1000 iterations). This behavior was consistent across three random seeds and both SMALL and MEDIUM settings.
>
> **Conclusion:** The system does *not* require full retraining when costs change. Updating the cost predictor is sufficient.
>
> Plots showing the evolution of cost and reward over time have been added to the revised paper (not included here due to rebuttal format constraints).
>
> ## How Dynamic Library expands the reachable set?
>
> It is true that, by definition, any intermediate molecule is reachable via some trajectory. However, once a molecule is added to the dynamic library, it becomes available as a reactant. This enables synthesis paths (trajectories) to become non-linear, as a reactant can now be a previously synthesized molecule—rather than being limited to the pre-defined reactant pool. This is illustrated in **Figure 3**: the molecule highlighted in red is itself a synthesis tree and could not have been reused as a reactant without the Dynamic Library. More formally, SCENT without the dynamic library is restricted to synthesis trees in which each node has at most one non-leaf child. Under this constraint, not all molecules are reachable. To experimentally validate that, we randomly generated 10000 full synthesis trees (balanced binary trees) using fragments and templates from MEDIUM settings, and 43% of the obtained molecules were observed to be non-decomposable with standard tree-like trajectory, confirming our claim.
>
> ## Dynamic Library hyperparameters sensitivity
>
> We conducted an extensive ablation study to assess the sensitivity of SCENT to hyperparameter choices across all components. Detailed results are provided in the response to Reviewer QKfB (Section *Hyperparameters*). We find that reducing the number of intermediates can slightly improve performance, while changing the update frequency of the dynamic pool has negligible effect. Overall, SCENT exhibits strong robustness to the hyperparameters of the Dynamic Library.
>
> ## How are the reaction yields obtained?
>
> Thank you for noting the imprecision in Appendix C. For reactions where sufficient in-house experimental data was available (amide coupling, nucleophilic additions to isocyanates, Suzuki reaction, Buchwald–Hartwig coupling, Sonogashira coupling, and azide–alkyne cycloaddition), yields were calculated as the average of all recorded experimental yields corresponding to each SMARTS-based reaction template. For the remaining reactions in the dataset, yield estimates were derived from the average of reported literature yields for reactions matching the respective SMARTS templates.
>
> ## Constant yields
>
> We fully acknowledge the reviewer's concern that our yield assignment method is simplified. However, in our view, this simplification is necessary, as no more accurate and universally applicable yield estimation methods currently exist [1]. Importantly, our cost‑biasing framework is fully modular and can accommodate **any recursively defined cost estimation**, including those with sophisticated, laboratory-specific yield‑prediction models.
>
> [1] “When Yield Prediction Does Not Yield Prediction: An Overview of the Current Challenges”
>
> ## Solvents and reagents are ignored
>
> Including solvent and reagent costs in template cost estimation is a valuable suggestion and may be important in practical applications. However, accurately estimating these costs is nontrivial, as they are often lab-specific and less standardized than building block prices. In our work, we focus on building blocks, which typically dominate overall cost, and adopt a simplified approximation that excludes solvents and reagents. While this may reduce accuracy for any specific lab setting, it offers a more robust and broadly applicable estimate.
>
> Importantly, our framework is flexible: more detailed or lab-specific cost models, including those that incorporate reagent pricing, can be readily integrated. We view this as a promising direction for future work.
>
> ## Pathways Analysis
>
> To visualize the cost guidance in practice, we gathered training trajectories from SCENT (C) and SCENT (w/o C) that lead to the same high-rewarded molecule in the SMALL setting. We added them to revised paper, and included them in a form of Reaction SMILES in response to Reviewer **ptir** under *Pathways Examples* section.
>
> We observe that, on average, cost-guided trajectories are ~10% cheaper. We selected a few example trajectory pairs and noticed that the cost reduction usually occurs by changing the order of the reactions: introducing expensive fragments later in the synthesis path, which decreases the product losses due to imperfect yields, and as a result increases the cost-efficiency. In **Example 1** (response to Reviewer **ptir** under *Pathways Examples***)**, SCENT (C) chooses the most expensive building block in the sequence (OB(O)c1ccc2[nH]ccc2c1) at the beginning of the synthesis, while SCENT (C) utilizes it in the second step, effectively reducing the synthesis cost by almost 28%. Cost-guided model also occasionally prefers cheaper fragments, e.g., *Brc1c[nH]cn1 instead of Ic1c[nH]cn1.* It is worth emphasizing that the fragments in the SMALL setting are already low cost, which is the reason why most gains are observed due to reaction order swapping.
>
> Similarly, we compared the Dynamic Library, SCENT (C+D), to SCENT (C), and observed that on average the trajectories are 12% shorter and 6% cheaper. We selected few examples that visualized the compressed trajectories. We have found that the Dynamic Library enables the generation of convergent synthetic pathways (Example 2), which are responsible for the reduction of the synthesis cost. The application of the convergent synthetic strategy decreases the longest linear sequence (LLS) of steps, thereby increasing the overall yield.
>
> In the MEDIUM settings, the SCENT variations diverged early on, so the set of shared molecules is very limited. Therefore, we matched high-rewarded molecules from SCENT (C) to closest high-rewarded molecule from SCENT (w/o C) with Tanimoto similarity > 0.6. We observed that, on average, the cost-guided trajectories are 20% cheaper. Thorough investigation of few selected trajectories pairs revealed that, differently from SMALL setting, in MEDIUM setting, the cost reduction comes mostly from the selection of cheaper fragments (Example 3).
>
> ## No limitation section (and future work)
>
> We apologize for the lack of limitation section, we are going to include the following limitation summary in the revised paper:
>
> “While our study demonstrates the effectiveness of SCENT in generating cost-efficient and synthesizable molecules, several limitations remain.
>
> First, the synthesis cost estimates are simplified and heuristic, omitting factors such as solvent usage, purification steps, hazardous reagents, reaction conditions, and scalability. Consequently, the current cost model should be regarded as a first-order approximation. More accurate estimation would require lab-specific calibration, including robust yield prediction models and vendor-specific building block pricing.
>
> SCENT’s performance varies across different biological targets, indicating that its effectiveness may depend on task-specific factors. Identifying and addressing these bottlenecks is an important direction for future work. Key areas of improvement include refining the exploitation behavior of the Dynamic Library, and enhancing the generalizability of the cost predictor model used in Synthesis Cost Guidance. Additionally, we do not assess performance as the number of reaction templates increases, an important scalability consideration, as this expansion significantly enlarges the action space and introduces additional learning challenges.
>
> Our framework emphasizes synthesizability and cost, but does not incorporate toxicity or broader ADMET properties, which are critical for downstream drug development. Finally, we acknowledge the absence of wet-lab validation, which remains the definitive benchmark for evaluating the practical utility of our approach. We leave this to future work.”
>
> ## Unrealistic one-step fragments
>
> The reaction Dynamic Library does take into account the cost of multi-step synthesis. Each intermediate molecule is assigned the lowest-cost synthesis path identified at the time it is added to the building block set. We acknowledge that this important detail was not clearly stated in the original manuscript, and we are grateful for the reviewer’s observation. We will clarify this point in the revised version of the paper.

---

> ### Comment · Reviewer_9Lb3 · 2025-08-08
>
> I appreciate the detailed authors' responses, and after reading the replies to the other reviewers' comments, I believe the changes, particularly those regarding the dynamic library hyperparameter sensitivity and warm-start, will increase the quality of the draft. I maintain my positive outlook on the paper and my positive recommendation remains unchanged. Thank you.

---

### Decision · Program_Chairs · 2025-09-17

**Decision:**

Accept (poster)

**Comment:**

This paper addresses the problem of molecular generation and proposes a new method based on the recently introduced GFlowNets, while also addressing their drawbacks, namely, generating diverse molecules while reducing synthesis cost.

The paper is overall well written and well organized. The proposed method has several strengths: it incorporates synthesis cost into the generation process, which is a practically relevant contribution; it leverages a dynamic library for more effective molecule generation; and it provides a thorough empirical evaluation.

The reviewers raised several concerns, such as the estimation of synthesis cost, evaluation of scalability, and the benefits of using the dynamic library. However, the authors responded well in their rebuttal, and all reviewers, except for one who remained unresponsive, expressed satisfaction and are positive about acceptance.

In my view, one weakness is that the proposed method is presented by describing each component in a straightforward manner. If Section 3 included a theoretical analysis or discussion of the properties of these individual components, the paper would be of even higher quality.

Overall, I believe this paper makes a strong contribution and I recommend its acceptance.